



# Graph Neural Operator for windfarm wake flow

Jens Peter Schøler[1], Frederik Peder Weilmann Rasmussen[1], Julian Quick[1], and Pierre-Elouan Réthoré[1]

[1]DTU Wind and Energy Systems, Frederiksborgvej 399, 4000 Roskilde

**Correspondence:** Jens Peter Schøler (jpsch@dtu.dk)

**Abstract.** Wind farm flow simulations are computationally expensive. However, numerous simulations are often required in applications such as wind farm layout optimization or when considering multiple neighboring farms, which motivates the development of data-driven surrogate models. However, most existing approaches rely on classical superposition principles, which constrain their ability to capture nonlinear wake interactions. We propose a novel method that embeds a trainable and scalable superposition principle within a Graph Neural Operator (GNO) architecture.

The model consists of two sequential Graph Neural Network (GNN) layers: the first encodes turbine–turbine interactions into a latent representation, while the second combines these latent turbine states to predict the wind speed at a desired location. The GNO is trained on a large dataset of simulated wind farms and achieves a low prediction error, with an RMSE of $0.353\,\mathrm{ms}^{-1}$ and a MAPE of $0.938\%$ on an unseen test dataset.

The GNO accurately identifies regions of strong wake interaction, although the spatial extent of wakes is slightly underestimated in cases with pronounced wake effects. Overall, the proposed GNO represents a significant advancement in data-driven wind farm flow surrogates, introducing a new conceptual framework inspired by established engineering wake modeling principles.



# 1 Introduction

Wind farm flow has been studied rigorously since engineers placed multiple turbines together in farms. The phenomenon of wind turbine-induced wakes is one of the most extensively studied subjects in wind energy, see, e.g., (Göçmen et al., 2016; Porté-Agel et al., 2020). Nevertheless, accurately representing the complex interactions among multiple wakes remains an active research challenge. In classical engineering wind farm flow models, the total flow field is obtained by calculating the operating state of each turbine and determining its corresponding wake contribution. The combined wind farm flow is then found through wake superposition. This step forms the core of most engineering models and significantly contributes to their overall performance.

Traditionally, wake superposition is performed using simple algebraic formulations based on the velocity deficit, either linearly (Lissaman, 1979; Niayifar and Porté-Agel, 2016) or quadratically (Katic et al., 1987; Voutsinas et al., 1990). While these formulations are computationally efficient, they rely on strong simplifying assumptions, in particular that wake interactions can be represented as additive. This means that critical physical processes such as wake mixing, entrainment, and interactions with the atmospheric boundary layer (Porté-Agel et al., 2020) are not adequately captured, which can lead to significant errors in the predicted farm flow. Recent studies have proposed modified superposition methods designed to enhance the physical realism of these models. The momentum-conserving superposition model by Zong and Porté-Agel (2020) introduces a weighted sum based on convective wind speeds, while the cumulative wake summation method by Bastankhah et al. (2021) enforces approximate mass and momentum conservation using an altered method of wake addition. These developments represent progress toward more consistent formulations, but they remain limited by their algebraic structure and cannot fully capture the nonlinear nature of wake interactions.

To overcome these limitations, more flexible and expressive operator formulations are needed, ones that can represent the inherently nonlinear and spatially coupled flow interactions occurring within wind farms. Machine learning offers a promising framework for this. Data-driven models can learn such complex wake interactions directly from data, without relying on restrictive analytical assumptions. In this context, graph learning provides a particularly suitable approach. By representing turbines as nodes and their aerodynamic couplings as edges, message-passing Graph Neural Networks (GNNs) can learn to propagate and combine wake information across the wind farm. This effectively generalizes the traditional wake superposition step into a learned, nonlinear operator that can capture the complex flow physics governing wind farm behavior.

GNNs have been successfully applied to similar problems in the past. Park and Park (2019) demonstrated a Physics-induced Graph Neural Network (PGNN) as an accurate and generalizable surrogate model for wind farm power estimation. Their method embeds engineering models into the network to learn physically plausible interactions, which they validated by applying it to a Wind Farm Layout Optimization (WFLO) problem. Bleeg (2020) presented a GNN trained on simulated Reynolds-Averaged Navier–Stokes (RANS) data capable of accounting for wake losses within a wind farm. Yu et al. (2020) trained a GNN using measurement data to superpose temporal states, although its applicability was limited to a single wind farm





at a time. Ødegaard Bentsen et al. (2022) employed a Graph Attention Network (GAT) to predict individual turbine power production based on engineering-model data.

Duthé et al. (2023); de Santos et al. (2024) trained GNNs on data generated from engineering models capable of predicting loads and power. Duthé et al. (2024) further developed this model and employed transfer learning to enhance data fidelity using a limited amount of mid-fidelity data from Dynamiks, a further development of HAWC2Farm (Liew et al., 2023), which implements the Dynamic Wake Meandering approach. Li et al. (2024) trained a graph transformer model to predict farm-level power and applied it to a static yaw optimization task.

Li et al. (2022) proposed a unique type of GNN for wake flow prediction, leveraging the GNN framework to predict the flow behind a single turbine. Their configuration resembles that of Convolutional Neural Networks (CNNs) but uses the flexibility of graphs to operate on the unevenly distributed RANS mesh. They employed layered Graph SAmple and aggreGatE (GraphSAGE) blocks to sample neighborhoods and propagate information efficiently throughout the domain. The model by Li et al. (2022) stands out as the only one that attempts to predict the flow field within the domain, rather than solely at the turbine locations. However, since their model predicts the flow only behind a single turbine, it still relies on classical wake superposition methods to reconstruct the overall flow of the wind farm.

In this work, a new Graph Neural Operator (GNO) model is proposed, capable of predicting the flow over an entire wind farm, not only at the individual turbines. The model is formulated based on the theoretical foundation introduced by Seidman et al. (2022) within the Nonlinear Manifold Decoders (NOMAD) framework. This approach allows the model to learn continuous flow representations in a physics-consistent manner while retaining the flexibility of graph-based learning. The development of the proposed GNO constitutes the main contribution of this paper. The GNO implementation is inspired by classical engineering models of wind farms, designed to predict spatially continuous flow fields across the entire domain. This extends the predictive capability beyond turbine-level quantities, enabling direct inference of flow fields from graph representations. Additionally, the GNO is rigorously tested to assess its capability.

Furthermore, a secondary contribution is the development of a data generation pipeline that combines state-of-the-art engineering models with stochastic sampling to produce physically realistic and diverse training data.

The structure of the article is as follows: section 2 introduces the employed methods and is divided into four subsections, concerning data generation, graphs, the GNO, and performance metrics. section 3 consists of two parts: In subsection 3.1 the results of a grid search to determine suitable hyperparameters are introduced, and in subsection 3.2 the best performing model is rigorously tested and discussed. Finally, in section 4 conclusions and suggestions for future work are summarized.





## 2 Methodology

Here, the methods used to construct the GNO, create training data, and evaluate the model are presented. In subsection 2.1, the data generation process is described, covering random layout generation, inflow generation, and wind farm flow simulation. A general introduction to graphs is provided in subsection 2.2, and subsequently, the proposed GNO is introduced in subsection 2.3, including an overview of the neural network methods used to construct it. Finally, in subsection 2.4, the methods for training and evaluation are described, along with the introduction of the performance metrics used.

### 2.1 Data generation

To ensure the dataset includes a representative subset of wind farm layouts and inflow conditions, these scenarios are procedurally generated. In total, 3,570 unique layouts are generated. Ten inflow conditions per layout were chosen to improve the neural network's ability to learn the correlation between the inflow conditions and wind farm wake deficit. 2,072 of the generated layouts are used for training, 500 are used for validation during training, and 998 layouts are set aside for testing. That means that for training 20,720 data points are considered, 5,000 for validation and 9,980 for testing.

#### *Layout generation*

Plant Layout Generator (`PlayGen`) by Harrison-Atlas et al. (2024) is used to generate wind farm layouts. `PlayGen` creates four types of layouts: *Cluster*, which uses Poisson disc sampling to iteratively generate the wind farms by selecting an existing turbine, generating a random angle and distance, and place the new turbine if it satisfies the spacing constraints. *Single string* creates a linear turbine arrays with inserted breaks, applies cumulative correlated noise to y-coordinates, and rotates the entire string. *Parallel string* makes multiple linear strings with the same orientation and vertically offsets each string by 1 to 3.5 rotor diameters ($D$). Then it applies random horizontal shifts and rotates the full layout. *Multi string* distributes turbines across strings, each with a quasi-random independent orientation, and places them in a domain while checking for string spacing. Examples of these are plotted in Fig. 1 (a)-(d).

The layout type, number of wind turbines ($n_{\mathrm{wt}}$) and the turbine separation factors ($s_{\mathrm{wt}}$) are sampled according to the probabilities and distributions listed below:

- Farm type: Categorical distribution sampled with probabilities $P_{\mathrm{farm}}$

$$P_{\mathrm{farm}} = \begin{bmatrix} P_{\mathrm{cluster}} \\ P_{\mathrm{single\ string}} \\ P_{\mathrm{parallel\ string}} \\ P_{\mathrm{multiple\ string}} \end{bmatrix} = \begin{bmatrix} 0.40 \\ 0.20 \\ 0.20 \\ 0.20 \end{bmatrix}$$

- Number of wind turbines $n_{\mathrm{wt}} \in \mathbb{Z}$ sampled with a Truncated Normal distribution TN

$$n_{\mathrm{wt}} \sim \mathrm{TN}(\mu = 60,\ \sigma = 60,\ \delta_{\mathrm{low}} = 20,\ \delta_{\mathrm{high}} = 100)$$





– Wind turbine separation factor $s_{\mathrm{wt}} \in \mathbb{R}_{>0}$ sampled with a Truncated Normal distribution TN

$$s_{\mathrm{wt}} \sim \mathrm{TN}(\mu = 5D,\, \sigma = 3D,\, \delta_{\mathrm{low}} = 2D,\, \delta_{\mathrm{high}} = 8D)$$

where $P$ are probabilities and subscripts indicate the layout type considered, $\mu$ is the mean, $\sigma$ the standard deviation, $\delta_{\mathrm{low}}$ the lower cut off and $\delta_{\mathrm{high}}$ the upper cut off for the truncated normal distribution TN.

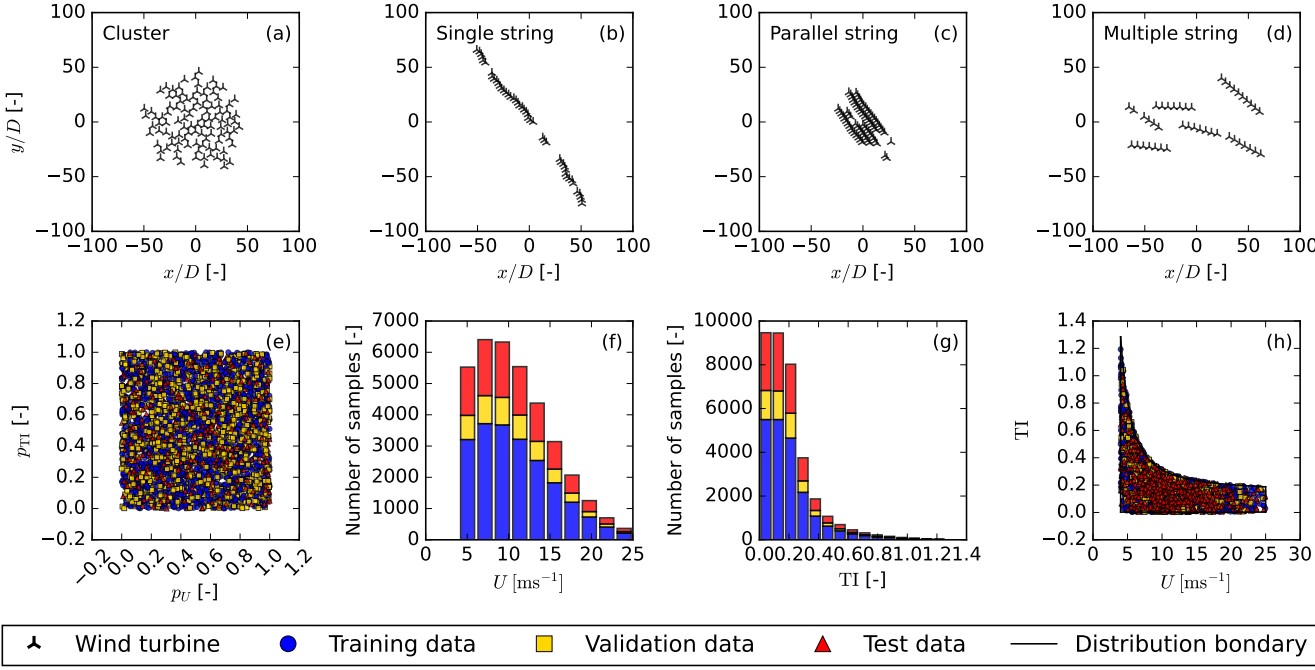

**Figure 1.** Procedurally generated data, (a-d) wind farm layouts using PLayGen. (e) Quasi random samples generated with the Sobol sequence. (f) $U$ distribution. (g) TI distribution. (h) Generated $U$ and TI with boundary.

*Inflow generation*

The inflow conditions required for the wind farm simulation are the free stream velocity ($U$) and ambient Turbulence Intensity

(TI); as these are naturally highly correlated, it is necessary to consider when the flow cases are generated. The methods of Dimitrov et al. (2018) are used to generate correlated inflow conditions. After Dimitrov et al. (2018) initially published, the IEC-61400-1 standard (IEC) has been updated with a slight change to the classifications of turbulence characteristics. Therefore, the new $\mathrm{A}^+$ class is used with reference turbulence intensity ($\mathrm{TI}_{\mathrm{ref,A}^+} = 0.18$). Ranges of the free stream velocity are based on the DTU-10-MW reference wind turbine (Bak et al., 2013) with rotor diameter ($D = 178.3$ m), cut-in ($U_{\mathrm{c,in}} = 4\ \mathrm{ms}^{-1}$), rated

($U_{\mathrm{rated}} = 11.4\ \mathrm{ms}^{-1}$) and cut-out ($U_{\mathrm{c,out}} = 25\ \mathrm{ms}^{-1}$) wind speeds. The power curve and coefficient of thrust ($C_{\mathrm{T}}$) curves of the DTU-10-MW are displayed in Fig. 2. The velocity standard deviation ($\sigma_u$) lower and upper bounds follow the expressions





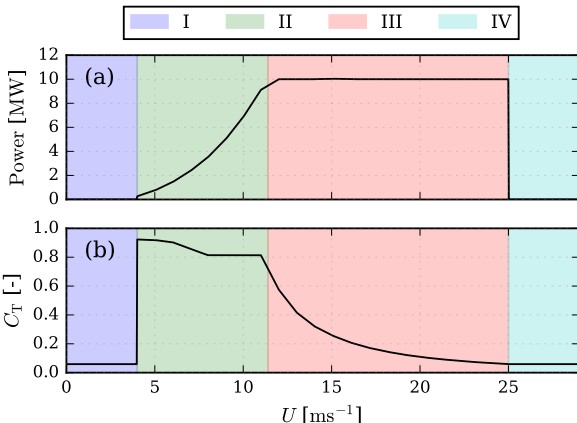

**Figure 2.** DTU-10-MW reference wind turbine (a) power curve and (b) $C_\mathrm{T}$ curve as implemented in `PyWake`. Operational stages I-IV are indicated with colors. I: below cut-in, II: Below rated power III: Rated power and IV: Above cut-out.

in Dimitrov et al. (2018, Tab. 1). The expressions for the inflow bounds are given in Eq. 1.

$$4\,\mathrm{ms}^{-1} \leq U \leq 25\,\mathrm{ms}^{-1} \tag{1a}$$

$$0.0025 \cdot U \leq \sigma_u \leq \mathrm{TI}_{\mathrm{ref,A^+}} \left( 6.8 + \frac{3U}{4} + 3 \left( \frac{10}{U} \right)^2 \right) \tag{1b}$$

$$[\mathrm{ms}^{-1}]$$

To generate the coupled flow cases the improved Sobol sequence by Joe and Kuo (2008) is used for Quasi-Monte-Carlo with correlated dimension pairs. Dimitrov et al. (2018) uses the Halton sequence, while de Santos et al. (2024) uses the improved Sobol sequence. Both methods were tested, and it was observed that the Sobol sequence better captures extreme values, so it was chosen for this work. In Fig. 1 (e) the components of quasi-random samples from the Sobol sequence are illustrated prior
to being projected through the target distribution. $p_U$ corresponding to the component of the sample used to generate $U$ and $p_{\mathrm{TI}}$ the component used to generate the ambient TI. In Fig. 1 (f) the resultant wind speed distribution is shown. The distribution was obtained by projecting the wind speed sample onto a Rayleigh distribution following IEC-61400-1 (IEC) and scaled with the range in Eq. 1a. To obtain the resultant TI distribution Eq. 1b is applied, and the turbulence standard deviation is converted to TI by the relation $\mathrm{TI} = \sigma_u/U$, the TI distribution is shown in Fig. 1 (g) as a histogram.

***Windfarm simulation***

The dataset is generated with the wind farm simulation tool `PyWake` (Pedersen et al., 2023). The wakes are modeled with the updated self-similar Gaussian single wake deficit model `NiayifarGaussianDeficit` by Bastankhah and Porté-Agel (2014); Niayifar and Porté-Agel (2016). This model was chosen due to its relative simplicity and its inflow TI dependent wake expansion. The model uses an adaptive wake growth rate ($k^*$) that is linearly fitted to the wind turbine inflow Turbulence





Intensity ($\text{TI}_{\text{wt}}$) immediately upstream of a given turbine. The authors fitted the model to a Large Eddy Simulation (LES), this work uses their default parameters $a_1$ and $a_2$.

$$\frac{\Delta u}{U} = \left(1 - \sqrt{1 - \frac{C_{\text{T}}}{8\left(\frac{k^*x}{D+\varepsilon_d}\right)^2}}\right)\exp\left(-\frac{1}{2\left(\frac{k^*x}{D+\varepsilon_d}\right)^2}\left(\frac{y}{D}\right)^2\right) \tag{2a}$$

$$\varepsilon_d = 0.2\sqrt{\beta_d}, \quad \beta_d = \frac{1}{2}\frac{1+\sqrt{1-C_{\text{T}}}}{\sqrt{1-C_{\text{T}}}}, \quad C_{\text{T}} < 0.9 \tag{2b}$$

$$k^* = a_1\text{TI}_{\text{wt}} + a_2, \quad a_1 = 0.3837, \quad a_2 = 0.003678 \tag{2c}$$

where $x$ is the streamwise direction, $\Delta u$ is the velocity deficit in the $x$-direction, $C_{\text{T}}$, is the coefficient of thrust, and $\varepsilon_d$ is a shape parameter offset dependent on $C_{\text{T}}$ and defined as in Eq. 2b.

To model added TI ($\Delta\text{TI}$), the `CrespoHernandez` model by Crespo and Hernández (1996) is chosen for its simplicity. The model depends on the induced velocity factor and the distance behind the turbine, as shown in Eq. 3.

$$\Delta\text{TI} = 0.73\, a_m^{0.8325}\, \text{TI}^{0.0325}\left(\frac{x}{D}\right)^{-0.32} \tag{3a}$$

$$a_m = 0.083\, C_{\text{T}}^3 + 0.0586\, C_{\text{T}}^2 + 0.2460\, C_{\text{T}} \tag{3b}$$

Where $a_m$ is the induced velocity factor estimated with an empirical polynomial fit of $C_{\text{T}}$ to address cases with $a_m \geq 0.5$ as described by Madsen et al. (2020).

To account for the effects of turbine induction, the updated self-similarity blockage model `SelfSimilarityDeficit2020` by Troldborg and Meyer Forsting (2017); Forsting et al. (2023) is used. The blockage model is used to calculate the blockage deficit produced by individual wind turbines ($\Delta u_b$). It is based on the observation that inductions are radially self-similar for upstream distances greater than one rotor radius ($R$). It consists of an axial and a radial-shaped function. The newer variation of the model has an updated linear induction zone half radius ($r_{1/2}$), which corrects the behavior of the turbine inductions in wind farm contexts.

$$\frac{\Delta u_b}{U} = a_0(x, C_{\text{T}})\nu(x)\text{sech}^\alpha\left(\beta_b\frac{r}{r_{1/2}(x)}\right), \tag{4a}$$

$$\nu(x) = \left(1 + \frac{x/R}{\sqrt{1+(x/R)^2}}\right), \tag{4b}$$

$$a_0(x, C_{\text{T}}) = \frac{1}{2}\left(1 - \sqrt{1 - \gamma(x, C_{\text{T}})\cdot C_{\text{T}}}\right), \tag{4c}$$

$$\frac{r_{1/2}(x)}{R} = \lambda\cdot(x/R) + \eta, \tag{4d}$$

$$\alpha = 8/9 \quad \beta_b = \sqrt{2} \quad \lambda = -0.672, \quad \eta = 0.4897$$





where $\nu$ is the centre-line induction and $a_0$ is the axial induction factor. The `SelfSimilarityDeficit2020` model introduced a $\gamma(x, C_T)$ function that gradually changes from a far-field expression to a near-field expression. Here, this is formulated as a function $\delta(x)$. The near- and far-field $\gamma$-functions are parameterized with $\overset{(i)}{c}_{nf}$ for the near-field $\gamma$ and $\overset{(i)}{c}_{ff}$ for the far-field $\gamma$.

$$
\gamma(x, C_T) = \Bigg\{ \delta(x) \cdot \left( \overset{(1)}{c}_{nf} \cdot \sin\left( \frac{C_T + \overset{(2)}{c}_{nf}}{\overset{(3)}{c}_{nf}} \right) + \overset{(4)}{c}_{nf} \right)
$$

$$
+ (1 - \delta(x)) \cdot \left( C_T^3 \overset{(1)}{c}_{ff} + C_T^2 \overset{(2)}{c}_{ff} + C_T \overset{(3)}{c}_{ff} + \overset{(4)}{c}_{ff} \right) \Bigg\} \tag{5a}
$$

$$
\overset{(1)}{c}_{nf} = -1.381, \qquad \overset{(2)}{c}_{nf} = 2.627,
$$

$$
\overset{(3)}{c}_{nf} = -1.524, \qquad \overset{(4)}{c}_{nf} = 1.336,
$$

$$
\overset{(1)}{c}_{ff} = -0.06489, \qquad \overset{(2)}{c}_{ff} = -0.4911,
$$

$$
\overset{(3)}{c}_{ff} = -0.1577, \qquad \overset{(4)}{c}_{ff} = 1.116, \tag{5b}
$$

$$
\delta(x) = \begin{cases} 1 & \text{for } x/R < -6 \\ \dfrac{|\nu(x) - \nu|}{\nu(-6) - \nu(-1)} & \text{for } -6 \leq x/R \leq -1 \\ 0 & \text{for } -1 < x/R \end{cases} \tag{5c}
$$

A linear sum is used for the wake summations. The velocity at turbine number $i$ is computed as

$$
u_i = U - \sum_{j=1}^{N_{up}} \Delta u_j - \sum_{k=1}^{N_{down}} \Delta u_{b,k} \tag{6}
$$

where $j$ sums across all turbines upstream of turbine $i$ and $k$ sums across all turbines downstream of turbine $i$.

Finally, to obtain the state of the wind farm, the `All2AllIterative` wind farm model is used, in which a local reference wind speed $u_{ref}$ is calculated per wind turbine. The inter-turbine effects are iteratively evaluated using fixed-point iteration until a convergence criteria is met. Using `All2AllIterative` is necessary when a blockage model is included because the turbines interact in both the upstream and downstream directions. After the dataset for this work was generated, a computationally lighter alternative wind farm model, the `PropagateUpDownIterative` model, was implemented in PyWake. This model can significantly accelerate dataset generation, and the authors encourage its adoption in future studies.

Once a $C_T$ is determined for all turbines, a flow map can be generated. The GNO is grid invariant, allowing for the strategic choice of a data coordinate system to capture interesting wake dynamics while avoiding redundant data from flow away from turbine wakes. Therefore, a bounding box is constructed to fit each farm. For each farm layout, the wind turbines with the minimum and maximum downstream $x$ and cross-stream $y$ coordinates are found and padded as follows:

– **Downstream:** ($x$-direction) an extension of $100D$ behind the most downstream turbine,





– **Upstream:** ($x$-direction) an extension of $10D$ in front of the most upstream turbine,

– **Cross-stream:** ($y$-direction) an extension of $5D$ on each side.

Fig. 3 illustrates the bounding box of a wind farm, and a farm downstream axis, $\bar{x}$, is introduced to simplify comparisons between different wind farms. The flow map is constructed to remain within the bounding box, at a single height, the turbine hub height ($z_{\text{hub}} = 119$ m). The flowmap uses an isotropic grid resolution of 3 points per $D$. For each layout and sampled inflow case, the computed effective turbine velocities and the velocities at the pre-determined grid locations are recorded.

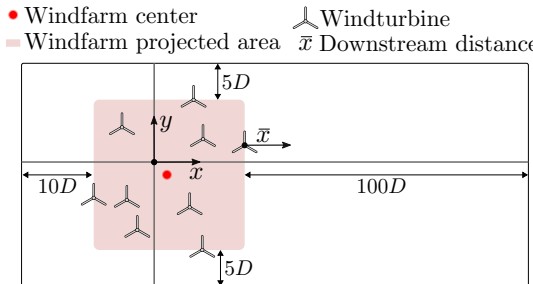

**Figure 3.** Adaptive computational grid for generating flow maps with `PyWake`.

**2.2   Graphs**

Graph theory was introduced by Euler (1741) to address the Königsberg bridge problem. Euler modeled land masses as nodes ($v$) and bridges as edges ($e$) in a graph. Although simple compared to modern graphs, this representation enabled Euler to mathematically prove the impossibility of solving the Königsberg bridge problem.

     Graphs are powerful tools for representing data. They consist of nodes (or vertices), edges, and optionally global features.
Nodes hold values representing information such as positions and states, while edges represent relationships or interactions between nodes and may also carry additional attributes, referred to as edge features. Global features, in turn, apply to the entire graph and are often used in graph classification tasks. In this work, global features are not included directly; instead, they are copied to each node as node features.

     The notation in this work uses both set theory and vector notation, depending on which is more appropriate in a given
context. In set notation, the nodes are represented by $V$ and the edges by $E$, with the graph represented as $G = (V, E)$. In vector notation, the nodes are denoted by $\boldsymbol{v}$, with individual vector elements represented as $v_i$; similarly, the edges are denoted by $\boldsymbol{e}$, with individual elements represented as $e_i$. Set theory is primarily used to express the cardinality of these sets. That is, the number of elements they contain, written as $|V|$ and $|E|$. The use of cardinality highlights that the numbers of nodes and edges are not fixed, which is a strength of graphs but also makes the notation more complicated.

In our implementation, two types of nodes are considered: wind turbine nodes ($V_{\text{wt}}$) and probe nodes ($V_{\text{p}}$), where

$$V_{\text{wt}} \subset V, \quad V_{\text{p}} \subset V \tag{7}$$





Wind turbine nodes coincide with the physical turbine positions in the farm, while probe nodes correspond to locations where flow predictions are desired, i.e., points in space at which the flow is evaluated. Similarly, the edges are separated into two inter-turbine edges ($E_{\mathrm{wt}}$) and those connecting wind turbines and nodes ($E_{\mathrm{p}}$), both a subset of $E$. The inter-turbine edges

are bidirectional, enabling message passing in both directions. This corresponds to solving for turbine thrust coefficients in traditional wake modeling, and the resulting latent space can be thought of as a high-dimensional representation of the turbine thrust states.

By contrast, edges connecting to probe nodes are unidirectional, with each probe node connected to and receiving from all wind turbine nodes. This corresponds to using a wake simulation result to predict a resulting flow map when the turbine thrust

states are already set. Probe node connectivity is straightforward because each probe connects to all wind turbines. Inter-turbine connectivity is more complex because it uses Delaunay triangulation to determine which turbines should be connected.

Several algorithms exist for constructing turbine graph connectivity. Duthé et al. (2023) investigated four graph connectivity schemes: Delaunay triangulation (Delaunay, 1934; O'Rourke, 1988; Fey and Lenssen, 2019), K-Nearest Neighbors (KNN), a radius-based method, and a fully connected scheme. They concluded that Delaunay triangulation provides the best balance

between accuracy and computational performance. Accordingly, we also adopt Delaunay triangulation in this work to derive inter-turbine connections.

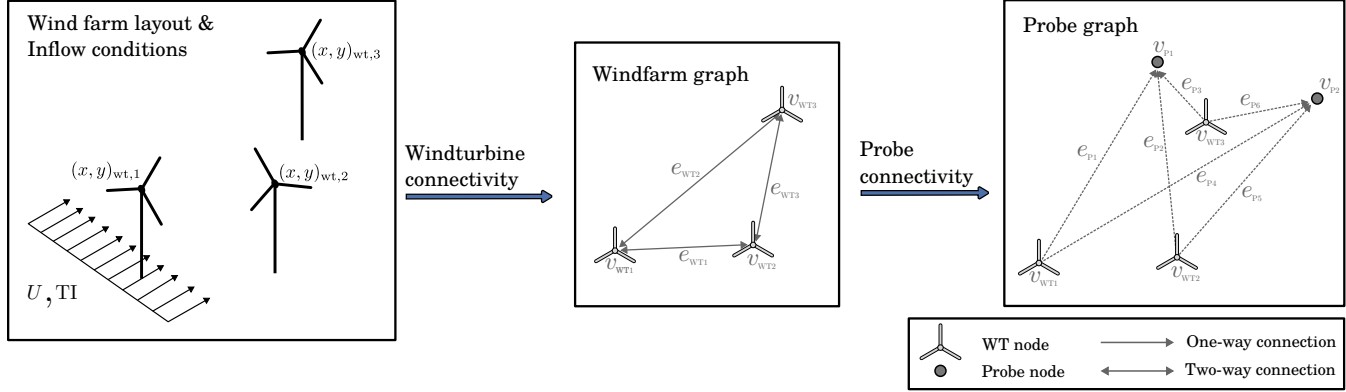

**Figure 4.** An example of constructing graph connectivity with three wind turbines and two probes, given a layout and inflow.

An example of the graph construction process is illustrated in Fig. 4, where the process of establishing the connectivity of the wind farm graph and the probe graph is separated into two stages. In our formulation, the edges include three features

($f_e = 3$): the position of the sending node in Cartesian coordinates (the streamwise $x$ and the cross-stream $y$ directions) and the

225 Euclidean distance between connected nodes. Initially, wind turbine nodes store two features ($f_v = 2$), namely the global wind speed $U$ and ambient TI. No unique information is stored directly on the nodes; instead, global features are initially seeded and replicated across the nodes as part of the processing. Since the desired output is the wind speed at probe nodes, this constitutes a node task.





## 2.3 Graph Neural Operator (GNO)

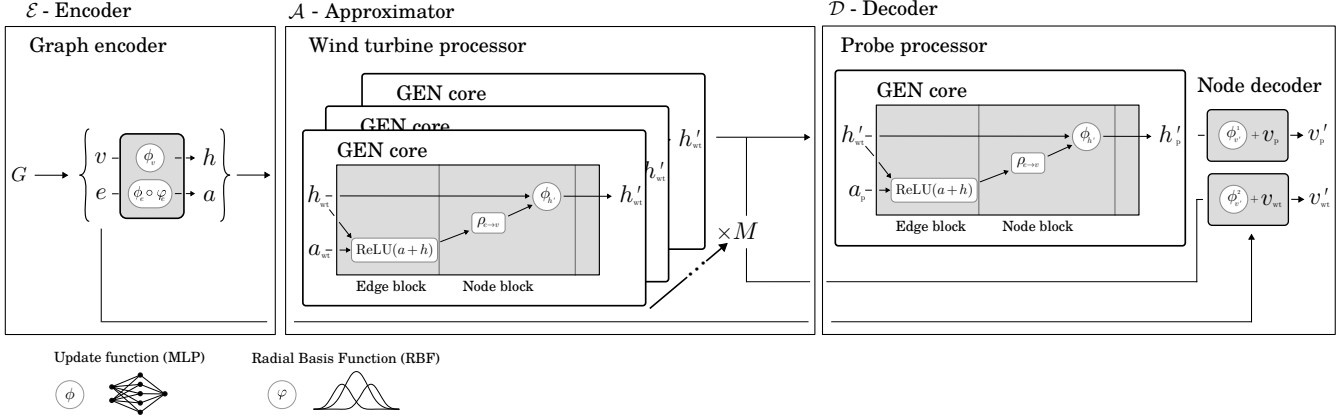

**Figure 5.** GNO network overview of its three main components: Encoder, Approximator and Decoder.

A Graph Neural Operator (GNO) is a special case of a Graph Neural Network (GNN). The GNO was first proposed by Li et al. (2020b) as a general method to solve Partial Differential Equations (PDEs). Later Sun et al. (2022) proposed the Deep Graph Operator Network (DeepGraphONet), combining the Deep Operator Network (DeepONet) by Lu et al. (2021) and Message Passing GNNs (Scarselli et al., 2009; Gilmer et al., 2017). In this work, the DeepGraphONet is combined with the considerations of Seidman et al. (2022) on Nonlinear Manifold Decoders (NOMAD) for operator learning. NOMAD generalizes neural operators to fit into the encoder-processor-decoder abstraction. They propose an operator $\mathcal{G}$ can be approximated as $\mathcal{F}$ using an Encoder $\mathcal{E}$, an Approximator $\mathcal{A}$, and a Decoder $\mathcal{D}$. The approximation $\mathcal{F}$ can be written as a composition of the model components using the composition operator $\circ$, where the rightmost function is applied first.

$$\mathcal{G} \approx \mathcal{F} = \mathcal{D} \circ \mathcal{A} \circ \mathcal{E} \tag{8}$$

The Encoder, Approximator, Decoder configuration is also a common abstraction in GNNs, see e.g. (Battaglia et al., 2018). In Fig. 5 an overview of the considered model is shown, while the initial two stages conceptually are equivalent to most GNNs the final decoder stage stands out as the decoder includes a variation of a readout that uses a separate set of featured probe edges $E_\mathrm{p}$. By including $E_\mathrm{p}$, the relative position of the turbines to the probe location is included during the readout. Additionally, due to the separation of probe node processing and the simplicity of establishing probe connectivity, probes can be created on the fly without re-establishing wind turbine connections i.e. $E_\mathrm{wt}$, nor re-processing the wind turbine nodes. Which means predicting in a flow field can be sped up compared to a fully integrated prediction scenario.

As previously discussed, the structure of the GNO mirrors the modeling flow of engineering-based wind farm models. In such models, a connectivity matrix is established based on the spatial layout of the turbines and the chosen wake interaction scheme. For instance, in fully coupled formulations, all turbines influence one another, and the initial state is defined by the undisturbed inflow conditions. Analogously, the GNO constructs a graph representation of the wind farm, which is then





encoded into an initial higher-order thrust state. The approximator within the GNO can be interpreted as analogous to the iterative update process used in conventional wake models to adjust individual turbine states. However, in the GNO framework, this computation is performed in a latent space rather than directly in the physical domain. Finally, the decoder evaluates the aggregated flow response of the wind farm, corresponding to the reconstruction of a flow field or flow map in traditional engineering approaches. These parallels are made explicit in Eq. 8, where the constitutive components of the GNO reflect the sequential structure of conventional wind farm modeling frameworks.

To train the GNO, a combination of different Python-based frameworks is used. The GNN components are constructed with the Jax (Bradbury et al., 2018) based libraries; Jraph (Godwin et al., 2020) for GNN abstractions, and Flax (Heek et al., 2024) for neural networks. Additionally, as the Jax ecosystem does not yet have a dedicated data pipeline, graph construction and subsequent data loading are handled using PyTorch Geometric (PyG) (Fey and Lenssen, 2019). For additional details on the data pipeline, see Appendix A.

### *Encoding*

The graph encoder layer $\mathcal{E}$ consists of parallel encoders for nodes and edges. The nodes are encoded with a Multilayer Perceptron (MLP), and the edges are encoded in two steps: first, they are pre-processed with Radial Basis Functions (RBFs), and then encoded to the target latent space with an MLP. It is well known that there are different flow regimes at various downstream distances from a turbine. RBFs are used to encourage the network to view different downstream distances as distinct regimes. In initial experimentation, it was found to improve training. The input edge and node data are projected into latent spaces of the same dimensionality, denoted as $Q$, ensuring dimensional consistency as the chosen approximator core, GEneralized Aggregation Network (GEN), requires equal latent-space dimensions for nodes and edges. This is because the node- and edge-features are added together in the latent-space.

The structure of an MLP is described in three stages: the input layer in Eq. 9a, the hidden layers in Eq. 9b, and the output layer in Eq. 9c.

$$\boldsymbol{\xi}^{(0)} = \psi\left(\boldsymbol{W}^{(0)\top}\boldsymbol{x} + \boldsymbol{b}^{(0)}\right) \tag{9a}$$

$$\boldsymbol{\xi}^{(l)} = \psi\left(\boldsymbol{W}^{(l)\top}\boldsymbol{\xi}^{(l-1)} + \boldsymbol{b}^{(l)}\right), \quad l = 1, 2, \ldots, L-1 \tag{9b}$$

$$\phi(\boldsymbol{x}) = \boldsymbol{W}^{(L)\top}\boldsymbol{\xi}^{(L-1)} + \boldsymbol{b}^{(L)} \tag{9c}$$

Here, $\phi(\boldsymbol{x})$ represents the overall MLP mapping for input vector $\boldsymbol{x}$, $\boldsymbol{\xi}^{(l)}$ denotes the hidden units at layer $l$, $\boldsymbol{W}^{(l)}$ and $\boldsymbol{b}^{(l)}$ are the weights and biases of the network, respectively, and $\psi$ is the activation function. In this work, the activation function is the Rectified Linear Unit (ReLU). The MLPs considered in this work operate in real space and are used to map between different dimensions in both the latent and observational spaces, i.e., $\phi : \mathbb{R}^{n \times c} \to \mathbb{R}^{n \times q} \quad \forall \; n, c, q \in \mathbb{N}$.

RBFs are used to map the initial edge features, here distances between wind turbines $i$ and $j$ ($d_{ij}$), into a higher-dimensional space. We employ $K = 9$ Gaussian basis functions to transform distances such that $\varphi : \mathbb{R}^1 \to \mathbb{R}^K$. The formulation of the RBF





is shown in Eq. 10.

$$\varphi(d_{ij}) = \exp\left(-\beta_{\text{RBF}}^{(k)}\left(d_{ij} - \mu_{\text{RBF}}^{(k)}\right)^2\right) \cdot \delta_c\left(d_{ij}\right),$$

$$k = 1, \ldots, K, \tag{10a}$$

$$\delta_c(d_{ij}) = \begin{cases} 0.5 \cdot \left[\cos\left(\dfrac{\pi d_{ij}}{d_c}\right) + 1\right], & \text{for } d_{ij} \leq d_c, \\ 0, & \text{for } d_{ij} > d_c, \end{cases} \tag{10b}$$

where $\beta_{\text{RBF}}^{(k)}$ and $\mu_{\text{RBF}}^{(k)}$ are trainable parameters defining the $k$-th basis function, and $\delta_c(d_{ij})$ is a cosine cut-off function with cut-off distance $d_c$, ensuring a gradual and consistent decay. The initial $\mu_{\text{RBF}}^{(k)}$ are linearly spaced between -1 and 1, while $\beta_{\text{RBF}}^{(k)}$ all are initialized based on the maximum range and amount of basis functions $\beta_{\text{RBF}}^{(k)} = \frac{K}{\max(d_{ij}) - \min(d_{ij})}$. The initial RBF functions are illustrated in Fig. 6, where symmetry is emphasized by matching colors but using different line styles.

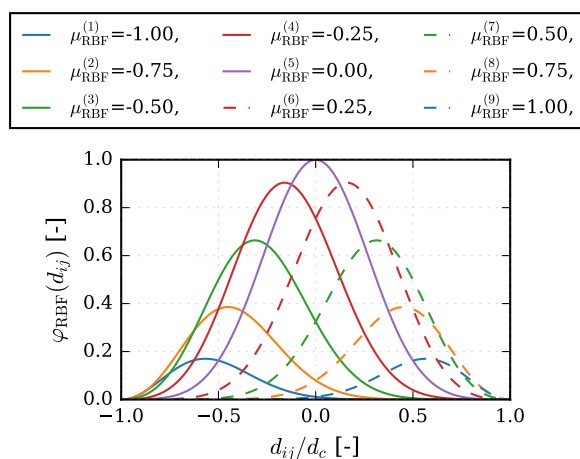

**Figure 6.** Initial RBF kernels for normalized distance and initial $\beta_{\text{RBF}}^{(k)} = 4.5$.

Combined, the encoding steps are summarized in Eq. 11,

$$\boldsymbol{h} = \phi_v(\boldsymbol{v}), \tag{11a}$$

$$\boldsymbol{a} = \left(\phi_e \circ \varphi_e\right)(\boldsymbol{e}), \tag{11b}$$

where $\boldsymbol{h}$ and $\boldsymbol{a}$ denote the latent-space representations of nodes and edges, respectively. $\varphi_e : \mathbb{R}^{|E| \times f_e} \rightarrow \mathbb{R}^{|V| \times f_e K}$ is the RBF function mapping each edge feature from the initial observational space to a $K$ dimensional RBF space, and $\phi_v : \mathbb{R}^{|V| \times f_v} \rightarrow \mathbb{R}^{|V| \times Q}$ and $\phi_e : \mathbb{R}^{|E| \times f_e K} \rightarrow \mathbb{R}^{|E| \times Q}$ are the node and edge encoding MLPs.



### *Approximator: GEneralized aggregation Network (GEN)*

The central component of the GNO is the approximator $\mathcal{A}$. It applies the GEN message passing algorithm with three message passing steps ($M = 3$) sequentially applied on the encoded latent space wind turbine nodes ($\boldsymbol{h}_{\text{wt}}$) using the latent space inter-turbine edges ($\boldsymbol{a}_{\text{wt}}$) to obtain the processed wind turbine nodes ($\boldsymbol{h'}_{\text{wt}}$).

In our work, the GEN message passing core proposed by Li et al. (2020a) is used. The methodology consists in the construction of the messages ($\boldsymbol{m}_{ij}$), the application of the edge-to-node aggregation function ($\rho_{e \to v}$), and the node update MLP ($\phi_v$),

$$\boldsymbol{m}_{ij}^{(m)} = \text{ReLU}\left(\boldsymbol{h}_j^{(m)} + \boldsymbol{a}_{ji}^{(m)}\right) + \varepsilon, \qquad j \in \mathcal{N}(i) \tag{12a}$$

$$\boldsymbol{h}_i^{(m+1)} = \phi_v^{(m)}\left(\boldsymbol{h}_i^{(m)} + \rho_{e \to v}\left(\boldsymbol{m}_{ij}^{(m)}\right)\right), \qquad m = 1, 2 \dots M \tag{12b}$$

where $m$ is the current message passing step, $i \in I$ is the index of a receiving node, with $I$ being an index set relating to all receiving nodes, correspondingly $j$ indicates a sending node, $\mathcal{N}(i)$ is a set of the nodes neighboring the node with index $i$, and $\varepsilon = 1 \times 10^{-6}$ is a small positive value added for numerical stability. In Fig. 5, the GEN core is visualized schematically. In this work, Softmax aggregation is used as $\rho_{e \to v}$, it uses Softmax to scale the latent space of each feature dimension set ($\hat{\boldsymbol{x}}_j \in \mathbb{R}^{|\mathcal{N}(i) \times 1|}$).

$$\rho_{e \to v} = \sum_{\hat{\boldsymbol{x}}_j \in \mathcal{X}} \frac{\exp(\hat{\boldsymbol{x}}_j)}{\sum_{\hat{\boldsymbol{x}}_r \in \mathcal{X}} \exp(\hat{\boldsymbol{x}}_r)} \cdot \hat{\boldsymbol{x}}_j \tag{13}$$

where $\mathcal{X}$ is a collection of neighborhood features to be aggregated independently.

### *Decoder*

In the first part of the Decoder stage ($\mathcal{D}$), an additional message passing step is performed using the processed wind turbine nodes $\boldsymbol{h'}_{\text{wt}}$ with the latent-space probe edges $\boldsymbol{a}_{\text{p}}$ to obtain processed probe nodes $\boldsymbol{h'}_{\text{p}}$ consisting of aggregated information from all the wind turbine nodes. In the final part of the decoder stage, two separate MLPs map the latent variables back to the observational space. A residual-network (ResNet) formulation is adopted, in which the free-stream velocity $U$ is added to the decoded node values as the final step during inference. Consequently, this residual connection is also taken into account during the evaluation of the loss function during training.

$$\boldsymbol{v'}_{\text{p}} = \phi_{v'}^{(1)}(\boldsymbol{h'}_{\text{p}}) + \boldsymbol{v}_{\text{p}}, \tag{14a}$$

$$\boldsymbol{v'}_{\text{wt}} = \phi_{v'}^{(2)}(\boldsymbol{h'}_{\text{wt}}) + \boldsymbol{v}_{\text{wt}}, \tag{14b}$$

Where $\phi_{v'}^{(1)}$ and $\phi_{v'}^{(2)}$ are the decoding MLPs, both mapping from the latent space to the output space, $\phi_{v'}^{(1)}, \phi_{v'}^{(2)} : \mathbb{R}^{|V_{\text{p}}| \times Q}, \mathbb{R}^{|V_{\text{wt}}| \times Q} \to \mathbb{R}^{|V}$. Here, $\boldsymbol{v}_{\text{p}}$, $\boldsymbol{h'}_{\text{p}}$, and $\boldsymbol{v'}_{\text{p}}$ denote the probe input nodes, hidden-state nodes, and predictions, respectively. Likewise, $\boldsymbol{v}_{\text{wt}}$, $\boldsymbol{h'}_{\text{wt}}$, and $\boldsymbol{v'}_{\text{wt}}$ correspond to the wind turbine input nodes, hidden-state nodes, and predictions.





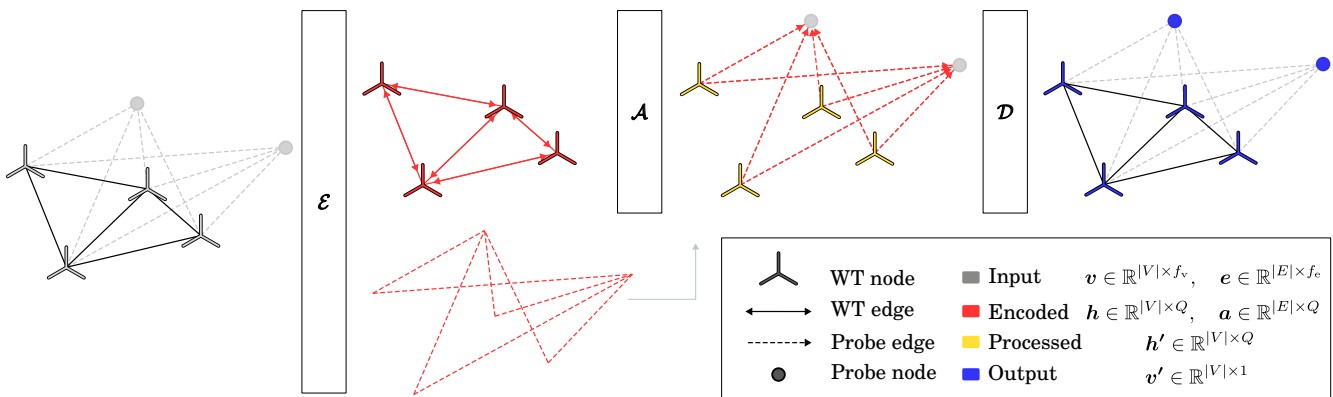

**Figure 7.** GNO Probe input, processing and output with model data flow from left to right.

The data flow through the GNO is shown in Fig. 7, where the components defined in Eq. 8 and described in this subsection are represented as boxes. The partially processed graph states are distinguished by color, providing a visual overview of the GNO architecture and its internal data transformations.

### *Regularization & Normalization*

With the layered nature of a GNO model the total amount of layers grow fast, with deep models it is advisable to include regularization and normalization to avoid overfitting. In this work, both the node- and edge features are scaled, meaning wind speeds, TI, and the positional information. Additionally, layer normalization and dropout regularization are used inside the GNO. For feature scaling min-max normalization is used.

$$\vartheta' = \frac{\vartheta}{\max(\vartheta) - \min(\vartheta)} \tag{15}$$

where $\vartheta$ is an unscaled feature and $\vartheta'$ is the respective scaled feature. The features are only scaled by the feature range as it allows ResNet application inside the scaled feature space, which makes model implementation easier.

To counter overfitting, dropout regularization, as implemented by Srivastava et al. (2014), has been incorporated into our MLP formulation. Dropout works by randomly omitting the output of some neurons during training, making the network more robust and reducing the risk of over-reliance on single neurons. Here, the Dropout method is used, along with the addition of inverse probability scaling, which means the entire Dropout layer can be ignored during inference. During training Dropout is applied both after Eq. 9a and Eq. 9b,

$$z_j^{(l)} \sim \text{Bernoulli}(P_{\text{D}}), \quad j = 1, 2, \ldots, N_a \tag{16a}$$

$$\tilde{\boldsymbol{\xi}}^{(l)} = \boldsymbol{z}^{(l)} \odot \frac{1}{1 - P_{\text{D}}} \cdot \boldsymbol{\xi}^{(l)}, \quad l = 1, 2, \ldots, L \tag{16b}$$





here $z^{(l)}$ is a Dropout mask consisting of zeros and ones, created with $\mathrm{Bernoulli}(P_\mathrm{D})$ the Bernoulli distribution given the Dropout probability, which here is chosen to be $P_\mathrm{D} = 0.1$, $N_a$ is the number of activations in layer $l$, the operator $\odot$ denotes the Hadamard element-wise product, and $\tilde{\boldsymbol{\xi}}^{(l)}$ is the resultant hidden state with Dropout applied.

A common technique to improve training stability and speed up training is batch normalization. However, as the trained GNO can handle graphs of any size, there is no fixed batch size, which makes batch statistics unreliable. Additionally, during training, graphs are batched together, and statistics across graphs are not desirable. For additional information about graph batching with GNOs, see Appendix A. Instead, Layer Normalization, as proposed by Ba et al. (2016), can be used. While 350 often considered less effective than batch normalization, layer normalization is fully compatible with GNNs. Ba et al. (2016) originally introduced it for MLPs and recurrent neural networks, applying it between hidden layers. In this work, since GNNs can be interpreted as compositions of MLP layers, layer normalization is applied only to the final layer of each MLP within the encoder $\mathcal{E}$ and approximator $\mathcal{A}$ stages, as well as during the message-passing step of the decoder $\mathcal{D}$ stage. It is not applied to the final node-decoder MLPs, where full expressive power at the output is desired. The formulation of layer normalization 355 used is shown in Eq. 17.

$$\mu_{\mathrm{LN}}^{(L)} = \frac{1}{H}\sum_{i=1}^{H}\xi_i^{(L)}, \qquad \sigma_{\mathrm{LN}}^{(L)} = \sqrt{\frac{1}{H}\sum_{i=1}^{H}\left(\xi_i^{(L)} - \mu_{\mathrm{LN}}^{(L)}\right)^2}, \tag{17a}$$

$$\bar{\boldsymbol{\xi}}^{(L)} = \boldsymbol{s}_{\mathrm{LN}} \odot \frac{\boldsymbol{\xi}^{(L)} - \mu_{\mathrm{LN}}^{(L)}}{\sigma_{\mathrm{LN}}^{(L)} + \varepsilon} + \boldsymbol{b}_{\mathrm{LN}}, \tag{17b}$$

Where $\mu_{\mathrm{LN}}^{(L)}$ and $\sigma_{\mathrm{LN}}^{(L)}$ denote the layer mean and standard deviation, $\xi_i^{(L)}$ is the $i$th activation at the last MLP layer $L$, and $\bar{\boldsymbol{\xi}}^{(L)}$ represents the layer-normalized hidden states, while $\boldsymbol{s}_{\mathrm{LN}}$ and $\boldsymbol{b}_{\mathrm{LN}}$ are trainable scale and bias parameters, and $\varepsilon = 1 \times 10^{-6}$ is 360 added for numerical stability.

### 2.4 Training and evaluation

The GNO is intended as a surrogate model for a regression-type task. Since a true model exists to learn from, the most straightforward training method is offline supervised training. To train the GNO, the High-Performance Computing (HPC) cluster Sophia (Technical University of Denmark, 2019) at DTU has been used. Nodes with `Quadro RTX 4000` GPU 365 acceleration were employed; these nodes have a limited runtime of 72 hours.

The optimization algorithm employed during training is `Adam` (Kingma and Ba, 2014). Adam is a variant of Stochastic Gradient Descent (SGD) that incorporates past gradients and a momentum term to compute parameter updates. Consequently, multiple hyperparameters control the relative weighting of these terms. In this work, the default momentum parameters are used. During training, the dataset is shuffled and passed through the training algorithm multiple times, one such pass is termed 370 an Epoch. In our implementation, a maximum of 3000 Epochs has been considered, although this limit has not been reached at any point. Instead, the limiting factor has been the 72-hour walltime. During training, the model is continuously evaluated





using the validation dataset. If the validation performance surpasses previous results at any time, the saved model weights and biases are updated. The validation evaluation is performed every fifth Epoch to save computational resources.

***Performance metrics***

Once a set of GNO weights are obtained through training, the resultant GNO model has to be tested. To accurately assess performance, various performance metrics are considered. Four statistical measures are considered: the Mean Absolute Error (MAE), the Mean Absolute Percentage Error (MAPE), the Mean Square Error (MSE), and the Root Mean Square Error (RMSE). They differ in the first two using an $l^1$-norm and last two using an $l^2$-norm. MSE and MAE are used to compare models, while MAPE and RMSE is chosen to evaluate the best model as they are more easily interpreted. The primary reason

for including multiple metrics is to facilitate cross-comparison with other works in the field, as there is no consensus on which metrics to use. The metrics considered here are shown in Eq. 18,

$$\mathrm{MAE} = \frac{1}{N} \|\boldsymbol{u} - \hat{\boldsymbol{u}}\|_1 \,, \tag{18a}$$

$$\mathrm{MAPE} = \frac{1}{N} \left\| \frac{\boldsymbol{u} - \hat{\boldsymbol{u}}}{\boldsymbol{U}} \right\|_1 \cdot 100\% \tag{18b}$$

$$\mathcal{L} = \mathrm{MSE} = \frac{1}{N} \|\boldsymbol{u} - \hat{\boldsymbol{u}}\|_2^2 \,, \quad \mathrm{RMSE} = \frac{1}{\sqrt{N}} \|\boldsymbol{u} - \hat{\boldsymbol{u}}\|_2 \tag{18c}$$

where $\boldsymbol{u}$ is the target velocities and $\hat{\boldsymbol{u}}$ is the predicted velocities, $\boldsymbol{U}$ is the corresponding inflow velocities, and $N$ is the number of observations. MSE is used as the loss function ($\mathcal{L}$) during training, as shown in Eq. 18c.

For an easy overview of how the errors vary, it is effective to estimate the Probability Density Function (PDF). To estimate the PDF, Kernel Density Estimation (KDE) is applied. The implementation of KDE uses a Gaussian kernel, and Scott's rule (Scott, 1992) to estimate the bandwidth $H$. $H$ governs the strength of the smoothing effect.

$$\hat{\varphi}(\zeta) = \frac{1}{NH\sqrt{2\pi}} \sum_{i=1}^{N} \exp\left( -\frac{1}{2}\left( \frac{\zeta - \zeta_i}{H} \right)^2 \right) \tag{19a}$$

$$H = \sigma_{\mathrm{kde}} N^{-1/5} \tag{19b}$$

Here $\zeta$ is a random input, $\sigma_{\mathrm{kde}}$ is the standard deviation of the sample data, and $\hat{\varphi}$ is the Gaussian kernel KDE function.

In some cases, it is practical to compare graphs by their relative size; a common approach to do so is to add up the size of their constituent parts. This can be done with different scaling factors and norms; however, the most straightforward and

common approach is a linear sum accounting for feature sizes. The cardinality of the graph tuple $|G|$ is therefore defined as:

$$|G| \equiv (|V_{\mathrm{wt}}| + |V_{\mathrm{p}}|) \cdot f_v + (|E_{\mathrm{wt}}| + |E_{\mathrm{p}}|) \cdot (1 + f_e) \tag{20}$$

Where the addition of 1 to edge set scaling $(1 + f_e)$ is added to account for the connectivity list in the practical implementation of a graph tuple.





### *IEA Wind 740-10-MW reference wind farm*

To evaluate the model under realistic conditions, the IEA Wind 740-10-MW reference wind farm is employed. This site features both a regular and an irregular turbine layout, corresponding respectively to a simple grid configuration and an optimized configuration designed to improve overall farm performance. Both layouts are tested under the same wind speed conditions as reported in the technical documentation by Kainz et al. (2024). The two layouts are shown in Fig. 8. As the GNO predicts wind speeds not power it is necessary to calculate power using the power curve of the DTU-10-MW illustrated in Fig. 2 a, as this is a static simulation this is a simple operation.

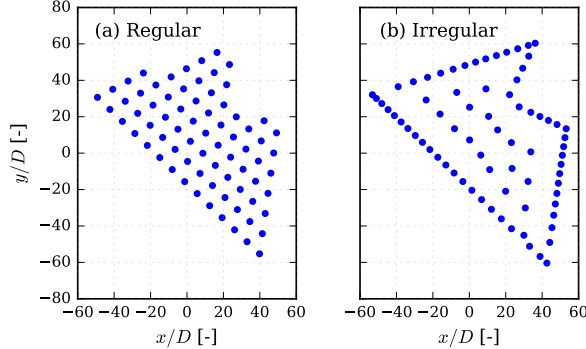

**Figure 8.** IEA Wind 740-10-MW reference wind farm (Kainz et al., 2024). (a) Regular layout pre-optimization and (b) Irregular optimized layout.





# 3 Results and Discussion

In this section the results are presented and discussed concurrently. Initially, a hyperparameter grid search study is presented, followed by the selection of the best model based on the validation data and its subsequent testing. The best performing model is investigated separately, and a series of predictions are made using random layouts of each considered layout type.

Both the probe node predictions and the wind turbine node predictions are investigated. Thereafter, a performance analysis is conducted, examining how errors are distributed across different input variables. Finally, the computational cost of the model is investigated and compared to the cost of PyWake.

## 3.1 Grid search

Neural network models consist of trainable parameters that are updated during training. In addition to these trainable parame-

415 ters, preset parameters known as *hyperparameters* also define the model. The hyperparameters can be divided into two groups: The ones defining the model architecture and those specifying the optimization configuration. In this work, five consecutive grid searches were performed, with the results from each used to guide the subsequent search.

For the model configuration, the tested hyperparameters are: the size of the latent dimension ($Q$), the number of layers in the internal MLPs ($L_{\mathrm{int}}$), the number of neurons per layer in the internal MLPs ($q_{\mathrm{int}}$), the number of hidden layers in the decoder

MLP ($L_{\mathrm{dec}}$), and the number of neurons per layer in the decoder ($q_{\mathrm{dec}}$). Table 1 lists the 11 considered model configurations, each assigned a letter ID.

Table 1 also includes the optimization configurations. The considered hyperparameters here are related to the Learning Rate (LR) schedule, which falls into two main categories: *constant* and *piecewise constant*. An additional category, *untriggered piecewise constant*, indicates that the LR remained at its initial value throughout optimization. This occurs because the max-

425 imum wall time has been reached, and the training is stopped. This occurs with configurations that have large input sizes, because each training step takes longer, resulting in fewer epochs being completed.

In the piecewise constant schedule, the current LR is divided by 10 when a trigger step is reached. The trigger steps themselves constitute another hyperparameter investigated in the grid search. Finally, two additional hyperparameters are considered: the number of probes per graph ($n_{\mathrm{p}}$) and the number of batched graphs ($n_G$). The batch includes an additional padding

graph consisting entirely of zeros; see Appendix A for further details on batching and padding graphs.

The different combinations of model and optimizer configurations were trained, and the results are listed in Tab. 2. Besides the model and optimizer IDs, a grid search number is provided. Three metrics are provided, training loss ($\mathrm{MSE}_{\mathrm{trn}}$), and two validation metrics ($\mathrm{MSE}_{\mathrm{val}}$) and ($\mathrm{MAE}_{\mathrm{val}}$). The entries in Tab. 2 have been ranked by the $\mathrm{MSE}_{\mathrm{val}}$ metric, as it serves as the selection criterion for the best overall model.

The best performing models in Tab. 2 show a high correlation between $\mathrm{MSE}_{\mathrm{val}}$ and $\mathrm{MAE}_{\mathrm{val}}$, which increases the confidence that the right model is chosen. The training loss $\mathrm{MSE}_{\mathrm{trn}}$ on the other hand is not as strongly correlated; this can partially be ascribed to the training dataset nodes being re-sampled during every epoch, which means there is a stochastic component to the $\mathrm{MSE}_{\mathrm{trn}}$ metric. By contrast, the validation metrics are deterministic and therefore better for comparison.





**Table 1.** Model and optimizer configurations used in the grid search. † indicates that the learning rate schedule was not triggered during training.

**Model configurations**

| ID | Latent dim. $Q$ | $L_{\text{int}}$ | $q_{\text{int}}$ | $L_{\text{dec}}$ | $q_{\text{dec}}$ |
|---|---|---|---|---|---|
| a | 150 | 2 | 100 | 3 | 150 |
| b | 150 | 2 | 100 | 3 | 250 |
| c | 150 | 2 | 200 | 3 | 150 |
| d | 150 | 2 | 200 | 3 | 250 |
| e | 250 | 2 | 100 | 3 | 150 |
| f | 250 | 2 | 100 | 3 | 250 |
| g | 250 | 2 | 200 | 3 | 150 |
| h | 250 | 2 | 200 | 3 | 250 |
| i | 100 | 3 | 75 | 3 | 250 |
| j | 100 | 2 | 250 | 3 | 350 |
| k | 50 | 3 | 350 | 4 | 350 |

**Optimizer configurations**

| ID | LR type | LR | Triggers | $n_{\text{p}}$ | $n_G$ |
|---|---|---|---|---|---|
| 1 | Piecewise constant† | 0.005 | [500, 1000] | 200 | 5 |
| 2 | Piecewise constant | 0.005 | [75, 150] | 200 | 5 |
| 3 | Constant | 0.001 | - | 200 | 5 |
| 4 | Piecewise constant | 0.001 | [75, 150] | 200 | 5 |
| 5 | Piecewise constant | 0.005 | [75, 150] | 300 | 4 |
| 6 | Piecewise constant | 0.001 | [75, 150] | 300 | 4 |
| 7 | Piecewise constant | 0.010 | [75, 150] | 200 | 5 |
| 8 | Piecewise constant† | 0.005 | [75, 150] | 500 | 2 |
| 9 | Piecewise constant† | 0.010 | [75, 150] | 500 | 2 |

In Tab. 3, the five best-performing models have been evaluated using the metrics in Eq. 18 and the unseen test dataset. All five

models perform similarly, indicating that, for the considered parameters, a minimum has been reached and that more substantial





**Table 2.** Trained models from the grid searches ranked by validation MSE$_{val}$ and lowest best metrics marked with b.

| Grid search | Model ID | Optimizer ID | MSE$_{trn}\times$ $10^{-3}$ | MSE$_{val}\times$ $10^{-3}$ | MAE$_{val}\times$ $10^{-3}$ |
|---|---|---|---|---|---|
| V | j | 8 | 0.018 | **0.018** | **1.266** |
| IV | k | 2 | 0.018 | 0.019 | 1.291 |
| IV | j | 2 | **0.016** | 0.021 | 1.412 |
| I | d | 1 | 0.021 | 0.024 | 1.538 |
| III | a | 6 | 0.020 | 0.026 | 1.978 |
| I | h | 1 | 0.018 | 0.026 | 1.482 |
| I | a | 1 | 0.024 | 0.027 | 1.788 |
| I | g | 1 | 0.021 | 0.027 | 1.692 |
| I | e | 1 | 0.027 | 0.027 | 2.073 |
| III | a | 5 | 0.029 | 0.027 | 1.916 |
| II | h | 2 | 0.019 | 0.028 | 1.707 |
| III | a | 4 | 0.021 | 0.031 | 2.211 |
| III | i | 6 | 0.021 | 0.032 | 2.054 |
| I | f | 1 | 0.026 | 0.032 | 1.899 |
| IV | f | 2 | 0.030 | 0.034 | 2.157 |
| V | k | 8 | 0.022 | 0.035 | 1.947 |
| IV | j | 7 | 0.021 | 0.035 | 1.967 |
| III | i | 4 | 0.022 | 0.040 | 2.804 |
| V | f | 9 | 0.036 | 0.046 | 1.973 |
| II | f | 2 | 0.025 | 0.046 | 2.341 |
| I | b | 1 | 0.025 | 0.057 | 2.610 |
| V | f | 8 | 0.028 | 0.062 | 2.201 |
| III | i | 2 | 0.026 | 0.062 | 3.938 |
| V | j | 9 | 0.023 | 0.063 | 2.208 |
| I | c | 1 | 0.020 | 0.075 | 2.582 |
| III | i | 5 | 0.023 | 0.088 | 4.807 |
| IV | f | 7 | 0.034 | 0.116 | 3.584 |
| III | a | 2 | 0.029 | 0.166 | 4.693 |
| V | k | 9 | 0.119 | 0.251 | 6.398 |
| II | f | 3 | 0.197 | 0.273 | 7.646 |
| IV | k | 7 | 0.196 | 0.275 | 7.799 |
| II | h | 3 | 0.198 | 0.281 | 7.115 |





**Table 3.** Test set error metrics for the five best performing models based on the grid search, with the best metrics marked in bold.

| Grid search | Model ID | Optimizer ID | MSE | MAE | RMSE | MAPE |
|---|---|---|---|---|---|---|
| V | j | 8 | **0.124** | 0.105 | **0.353** | 0.938 |
| IV | k | 2 | 0.127 | 0.106 | 0.356 | 0.957 |
| IV | j | 2 | 0.127 | **0.105** | 0.357 | **0.937** |
| I | d | 1 | 0.133 | 0.112 | 0.365 | 1.010 |
| III | a | 6 | 0.130 | 0.106 | 0.361 | 0.959 |

model changes are necessary to further improve performance. While the top model from the grid search in Tab. 2 maintains the best score in terms of MSE and RMSE, it is the third-best model that achieves the best MAE and MAPE. Interestingly, these two models share the same model configuration $j$, but differ in the optimizer configuration, with the variation arising from the input configurations of $n_{\mathrm{p}}$ and $n_G$. This could indicate that the input configuration is of less importance compared to the model 445 configuration.

### 3.2 Model performance

In this section, the performance of the best model from Tab. 2 is investigated. The testing is conducted in steps initially, where a random selection of each layout type is used to illustrate the model capabilities in the far wake at different downstream distances $\bar{x}$, see Fig. 3 for the definition of $\bar{x}$. In the second step, a more data-centric approach is taken by illustrating the 450 error statistics using the test set and analyzing the error with relation to different input types. Then, the model computational cost in terms of speed and memory is assessed, while varying the number of probe nodes to see the impact of graph size on computational cost.

#### *Predictions*

In Fig. 9 (a)-(d), velocity deficit ($\Delta u$) predictions made with the GNO are compared to the PyWake test data for a random layout 455 of each layout type. The predictions are illustrated for three downstream distances $\bar{x} = \{25, 50, 100\}\ D$, at three freestream wind speeds $U = \{6, 12, 18\}\ \mathrm{ms}^{-1}$ and TI = 5%. Unsurprisingly, the cluster farm with a number of wind turbines, $n_{\mathrm{wt}} = 90$, has the most significant wake effect but also the most smeared wake deficit, especially visible for the $U = 6\ \mathrm{ms}^{-1}$ scenario. The remaining layouts have fewer wind turbines, but also more structured layouts. They show clearer peaks and valleys, especially the single string layout. While the performance at $U = 6\ \mathrm{ms}^{-1}$ is satisfactory for all layouts, it becomes less accurate at higher 460 wind speeds. At $U = 12\ \mathrm{ms}^{-1}$, the model performs the worst, although it captures the shape of the velocity deficit; the scale could be improved. In Fig. 9, the velocity deficit error ranges between $\sim 2\%$ to $\sim 5\%$ relative to $U$.

As a secondary output, the model can predict the velocity at individual turbines. To illustrate this capability, Fig. 10 shows the maximum error observed at each turbine for the same layouts and inflow velocities as those presented in Fig. 9. In Fig. 10,







**Figure 9.** (a-d) Normalized velocity deficit predictions and targets at different wind farm downstream distances $\overline{x} \in \{25D, 50D, 100D\}$. (a) cluster, (b) single string, (c) multiple string and (d) parallel string.



WIND
ENERGY
SCIENCE
DISCUSSIONS

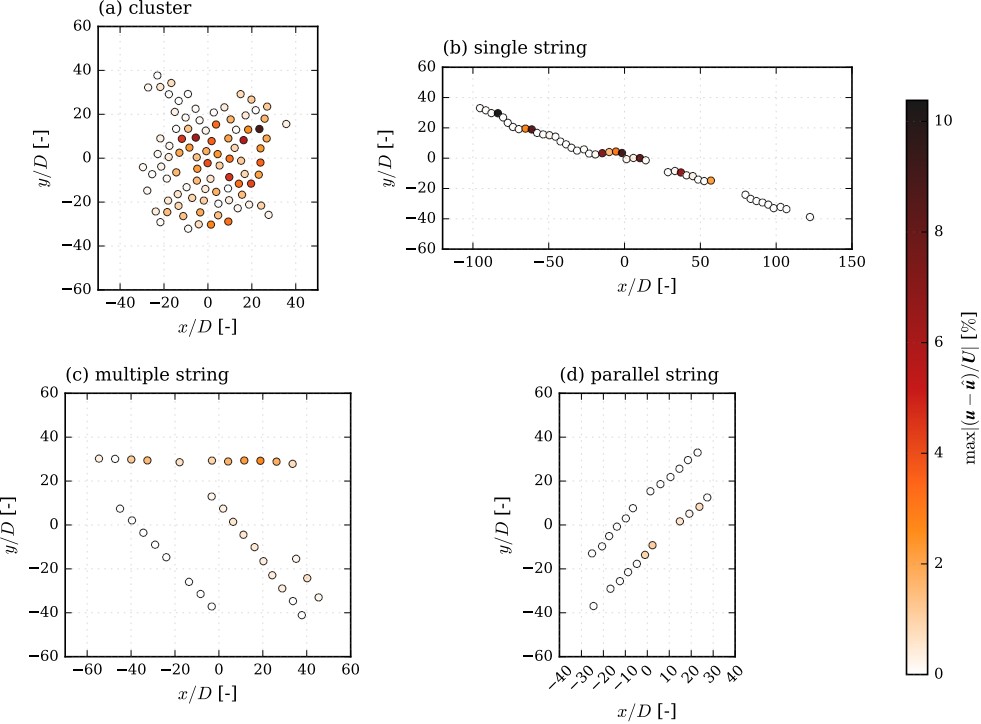

**Figure 10.** Maximum absolute errors at each wind turbine, for $\boldsymbol{U} = [6,\ 12,\ 18]^\top$ ms$^{-1}$

it can be seen that heavily waked turbines have the largest errors in Fig. 10 (c) and (d) where the turbines are not oriented in
a way that creates as deep and dense rows as in Fig. 10 (a) and (b), the wake effect is not as pronounced and the errors are
smaller.

### IEA Wind 740-10-MW reference wind farm

The reference wind farm consists of two layouts a regular and an irregular as described in subsection 2.4. For each layout, the
total farm power output at each wind direction is computed and visualized using a polar grid. As the GNO does not inherently
account for wind direction, this is achieved by translating and rotating the wind farm layout. In this way, each wind direction
is represented as an equivalent new configuration. Repeating this process across all directions enables the calculation of wind-
direction dependent farm power, without requiring the model to explicitly encode directional information. This approach is
feasible because the GNO has been trained on a large and diverse dataset comprising numerous wind farm configurations.
Consequently, it generalizes well across a wide range of geometric arrangements and inflow conditions. The wind-direction
dependent farm power for a turbulence intensity of $\text{TI} = 5\%$ are presented in Fig. 11.

As shown in Fig. 11a, the regular grid layout being non-optimized exhibits more pronounced internal wake effects and
therefore produces lower overall power. In contrast, the irregular layout in Fig. 11b shows reduced wake interactions and





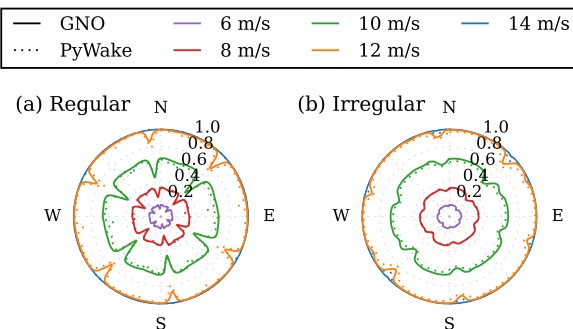

**Figure 11.** Normalized power production at different wind speeds and $\mathrm{TI} = 5\%$, illustrated as windroses for the IEA Wind 740-10-MW reference wind farm (Kainz et al., 2024). (a) Regular layout pre-optimization and (b) Irregular optimized layout.

higher power production across most wind directions. This result reflects the optimized nature of the irregular layout, which minimizes wake losses and enhances overall farm performance. The GNO reproduces the overall power trends well for both layouts, with slightly improved accuracy for the irregular configuration. This behaviour is consistent with previous observations that data-driven models tend to perform best under conditions similar to their training data, and where variability is lower. At a wind speed of $14 \ \mathrm{ms}^{-1}$, both farms operate close to rated power, leading to small differences in predicted output. Some discrepancies remain at lower wind speeds and for specific wind directions where wake effects are more pronounced. The GNO successfully identifies the regions of strongest wake interaction, although the scale of the effects is not reproduced exactly.

*Performance analysis*

To evaluate the performance of the model under different scenarios, predictions are made for all combinations of layouts and inflows in the test dataset. For each layout, 10,000 equally spaced probes are chosen, and an RMSE is calculated for each flow case. The resultant metrics are displayed in Fig. 12 with separate visualizations for each layout type.

In Fig. 12 (a), a KDE approximation of the RMSE PDFs is illustrated for each layout type. The PDFs show that the single- and multiple-string layouts have comparable PDFs, and they are simultaneously the best-performing layouts. The worst-performing layout type is the cluster, followed by the parallel string. To further investigate why that is the case, in Fig. 12 (b)-(e) bar plots of the errors are provided to investigate the effect of different aspects of the model inputs. Fig. 12 (b) shows a bar plot of the mean RMSE for different inflow wind speeds $U$. The four layouts share similar distributions with regard to $U$, all of them exhibiting the largest mean RMSE at $U = 12 \ \mathrm{ms}^{-1}$. The largest error occurring at $U = 12 \ \mathrm{ms}^{-1}$ is most likely related to the wind turbine $C_{\mathrm{T}}$-curve, as $12 \ \mathrm{ms}^{-1}$ is just past the steepest part of the curve in region III, see Fig. 2. This means that as turbines are affected by wakes inside the farm, the highest variability of $C_{\mathrm{T}}$ occurs at $12 \ \mathrm{ms}^{-1}$. Additionally, it is near the discontinuity of $C_{\mathrm{T}}$ when the rated wind speed is reached. To alleviate this, it is suggested that future implementations should rebalance the training dataset to proportionally include more flow cases right above the rated wind speed.

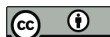
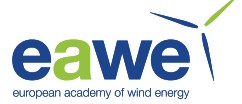


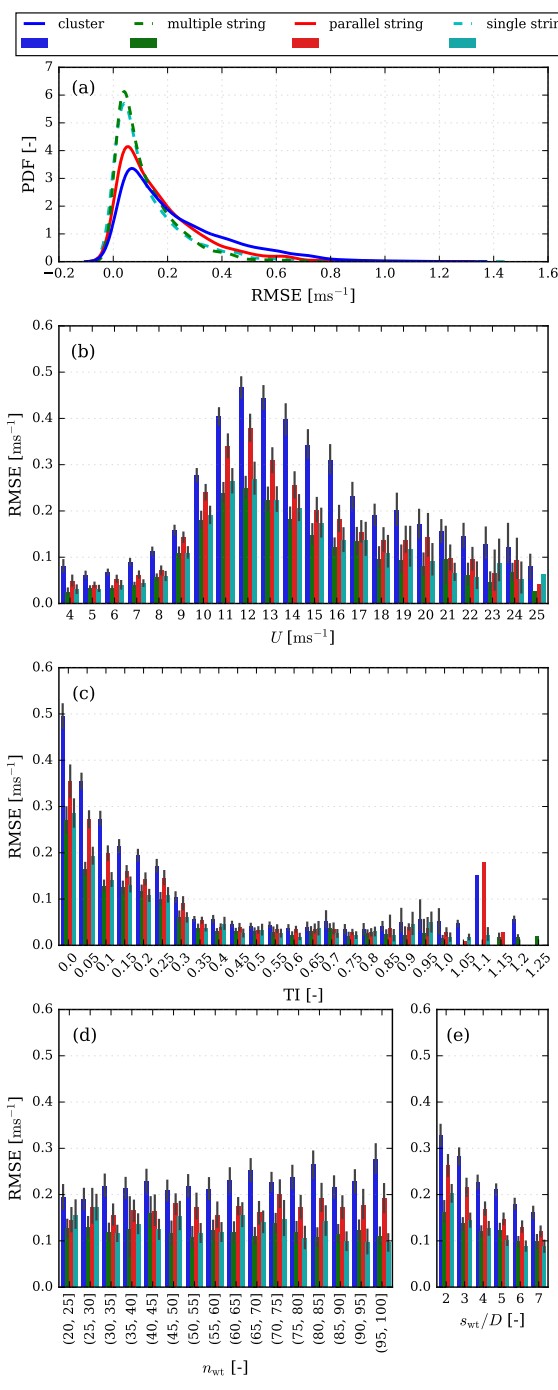

**Figure 12.** (a)-(e) RMSE metrics for different layout types. (a) RMSE PDFs estimated with KDE. Binned RMSE (b) at different free stream velocities $U$, (c) across different TI, (d) against the number of wind turbines $n_{\mathrm{wt}}$, and (e) with respect to the separation factor $s_{\mathrm{wt}}$





Fig. 12 (c) shows a bar plot of the mean RMSE as a function of TI. As expected, there is an inverse correlation between RMSE and TI. Lower TI indicates stronger wake effects, as the wake structure persists for longer, leading to a more complex farm flow and, consequently, higher errors. This occurs because turbulence breaks down wake structures and re-energizes the wind. At higher TI values, the RMSE increases again, but so do the associated error bars. At the highest TI levels, a few extreme cases have no error bars, as there is only a single sample and therefore no spread. The sparsity of high TI values arises

because the inflow cases were sampled to reflect realistic operating conditions rather than an even distribution of inflows. This is evident in Fig. 1 (g) and (h), where these extreme TI values are shown to be rare. Consequently, the limited number of samples for extreme TI values leads to the model being insufficiently trained for such scenarios, resulting in rising RMSE at the highest TI values. However, it is worth noting that these high TIs only occur at very low wind speeds and are extremely rare in the real world.

In Fig. 12 (d) and (e) the impact of the variables that govern the layout are investigated. These are the number of wind turbines ($n_{\mathrm{wt}}$) and the separation factor ($s_{\mathrm{wt}}$). All the layouts are strongly inversely correlated with $s_{\mathrm{wt}}$. The cluster and the parallel string layouts do show a slight correlation to $n_{\mathrm{wt}}$, but that is neither the case for the single string nor the multiple string layout types.

In summary, there is a strong correlation between the model accuracy, the inflow condition and the separation of the turbines.

Farms with fewer turbines and simpler wake interactions exhibit higher model accuracy, whereas larger dense farms, such as the cluster example in Fig. 9 (a), show greater errors due to increasingly complex and nonlinear inter-turbine interactions. The configuration of the farm influences this behavior: layouts with wider turbine spacings promote simpler flow patterns, while denser configurations, such as cluster and parallel string layouts, intensify wake interactions and adversely affect the GNO's predictive capability.

**Computational cost & Memory consumption**

For a surrogate model, it is important to understand the computational cost associated with its execution, as it should be faster than the actual model it replaces. As previously mentioned, the implementation is based on Jax, which enables Just-In-Time (JIT) compilation, making the GNO suitable for scenarios requiring multiple evaluations. For completeness, the GNO is evaluated in three configurations: (1) a pre-JIT-compiled state, which most closely reflects the intended real-world application;

(2) without JIT compilation; and (3) with JIT compilation, but performing only a single prediction. To encompass a wide range of graph sizes, the whole test set is used with a variable number of probe nodes, $n_{\mathrm{p}} \in \{1, 10, 100\} \cup \{1000, 2000, \ldots, 10000\}$.

The timings are compared against PyWake, where the number of probes is interpreted as the number of grid points in a flow map. All experiments are conducted using CPU resources for comparability, as PyWake does not currently support GPU acceleration. Timing results are reported in CPU hours (CPUh), as PyWake can be executed on a single CPU core, whereas the

surrogate model defaults to utilizing all available 32 cores. The CPU is not saturated for small and moderate graph sizes, putting the surrogate at a slight disadvantage. The results of this analysis are presented in Fig. 13. Fig. 13 (a) shows the overall timings, illustrating how costly it is to run the model without compilation and highlighting the significant upfront cost associated with



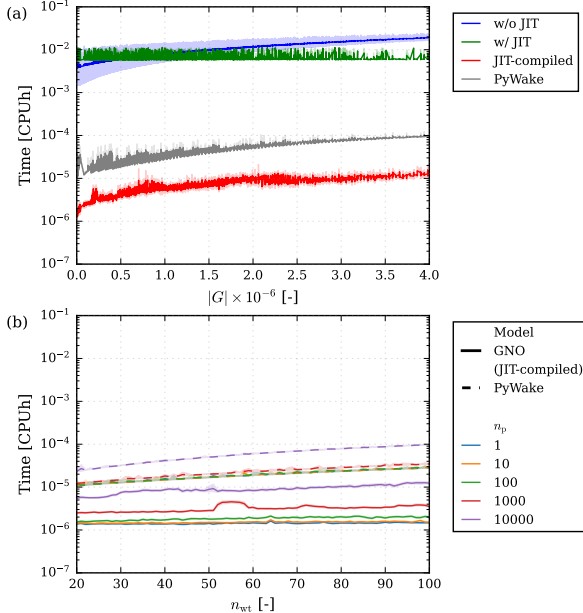

**Figure 13.** Model computational cost in terms of CPU hours. (a) Model cost of GNO in different JIT states compared to PyWake for varying graph sizes $|G|$. (b) Model computational cost in JIT-compiled state shown with different numbers of probes compared to PyWake for an increasing number of wind turbines.

compiling it. Once compiled, the results indicate that the computational cost of running the model increases with the graph size $|G|$, and that a performance improvement of approximately $\sim 10\times$ can be expected compared to PyWake.

To further investigate the GNO, each component described in subsection 2.3 was timed independently. The timings were measured both (1) in a pre-JIT-compiled state and (2) after JIT compilation. The results are shown in Fig. 14. The results show that, at larger graph sizes, prediction time is dominated by the encoding stage. In Fig. 14 (a), the computational cost increases almost linearly on the logarithmic axis, implying exponential growth in computational cost. This behavior is partly due to how the graphs were scaled by simply adding a large number of probes, which causes $|N_\mathrm{p}|$ and $|E_\mathrm{p}|$ to grow rapidly, while $|N_\mathrm{wt}|$

and $|E_\mathrm{wt}|$ remain unchanged. Consequently, the wind turbine interactions at the approximator stage are unaffected. Batching graphs is proposed for a more comprehensive analysis, but has not been investigated further.

    In Fig. 15, the memory consumption of the GNO is compared to the PyWake examples. As can be seen, there is no significant difference in consumed memory. However, at larger graph sizes, the memory consumption of the GNO starts growing more rapidly and, in some cases, overtakes the memory consumption of PyWake.

In summary, the GNO produces meaningful results that reflect the behavior of the flow around a given wind farm. As seen in Fig. 9, the accuracy increases with the distance behind the farm ($\bar{x}$). Therefore, a suitable application could be the assessment of long-distance wakes from neighboring farms. Furthermore, the GNO is particularly well-suited for this purpose, as the model is highly adaptable with respect to layout configurations. As a surrogate for engineering models, the GNO was found to predict



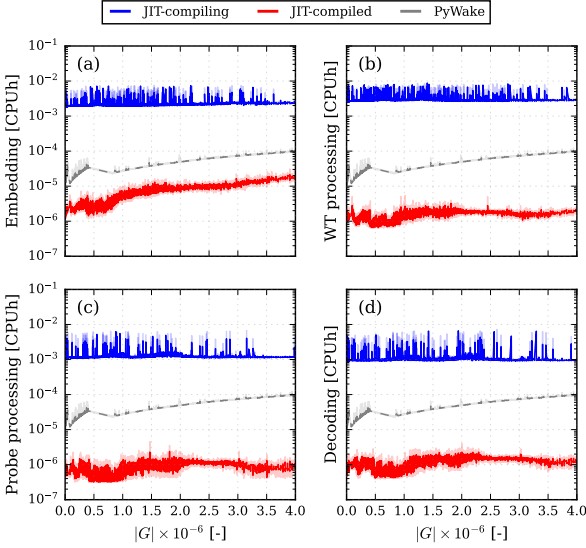

**Figure 14.** Timings of the GNO components. (a) Encoder $\mathcal{E}$, (b) Approximator $\mathcal{A}$, (c) Decoder $\mathcal{D}$ part 1: probe processing, and (d) Decoder $\mathcal{D}$ part 2: node decoding.

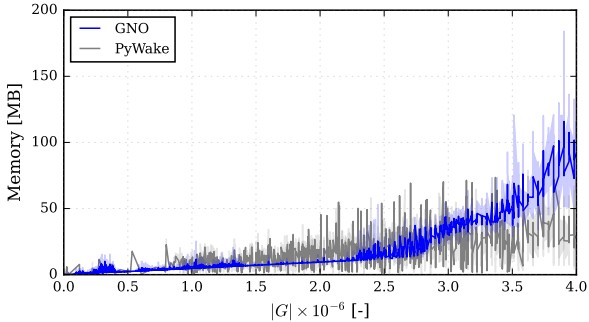

**Figure 15.** Memory consumption

approximately $\sim 10\times$ faster than PyWake. While this is an acceptable improvement, the speedup would increase significantly

if a higher-fidelity model, such as CFD, were used as the basis. For example, if RANS were considered, a speedup between $10^4$ and $10^5$ seems realistic given the relative cost difference between RANS and engineering models (van der Laan et al., 2015).



## 4 Conclusions

The GNO offers a fresh perspective on data-driven wind farm flow modeling. It has been established as a novel approach inspired by classic engineering models, demonstrating how the superposition principle can be integrated into the modeling process rather than being an afterthought, as is often the case in the established literature. Furthermore, the scalability and versatility of the graph-based approach have been shown to hold across highly varied layouts and inflow conditions, indicating that the model can perform well in a general sense.

In section 3, the performance of the GNO was evaluated. The results show that the model compares favorably in terms of computational cost and performs on par with PyWake in terms of memory consumption. The model was found to evaluate with a computational cost between $10^{-6}$ and $10^{-5}$ CPUh, equivalent to 3.6–36 ms on a single core, which is well within an acceptable range for practical applications. The GNO is less accurate than PyWake, which is expected, as a machine learning surrogate cannot fully replicate the fidelity of its source model. Nevertheless, the accuracy of the GNO remains reasonably high when evaluated on a previously unseen test dataset. An overall RMSE of $0.353 \, \mathrm{ms}^{-1}$ and a MAPE of $0.938\%$ were obtained. While these values constitute an acceptable error level, they do not provide the complete picture.

To gain deeper insight into the model performance, a more fine-grained assessment was conducted. First, the prediction error for representative cases was examined. This analysis demonstrated that the model accurately captures the wake shape, although some deviations persist in the predicted magnitude at medium wind speeds. A more comprehensive performance analysis revealed that the GNO performs best in scenarios with limited wake interactions. This trend is observed under inflow conditions with high TI, as well as at both low and high wind speeds. In contrast, performance deteriorates at medium wind speeds within the early part of turbine region III, where the high variability of $C_{\mathrm{T}}$ introduces additional complexity. Similarly, layouts with high turbine density, corresponding to a low separation factor, also result in reduced model accuracy. The impact of the total number of turbines was less pronounced, but a slight preference for smaller farms was observed for the *cluster* and *parallel string* layout types.

The GNO presented in this work constitutes a breakthrough in data-driven wind farm modeling. Nonetheless, several areas for improvement remain. Therefore, some suggestions are made for further work. The current dataset was created using engineering models, which, by design, apply a linear summation of wakes. Even though the `All2AllIterative` scheme introduces non-linear interactions between turbines, it still merges wakes linearly. The next logical step is to utilize higher-fidelity training data that incorporates turbine interactions directly into the model. The most suitable choice would likely be RANS, as its computational cost is significantly lower than that of other CFD methods. To further reduce the cost, a transfer-learning scheme could be introduced, combining engineering models and RANS, similar to the approach of Duthé et al. (2024).

The current GNO exclusively uses a GEN core, which, in combination with the Softmax aggregation, provides only a rudimentary attention mechanism. However, more powerful attention mechanisms exist in both GAT and full Transformer-based attention layers. Incorporating either of these into the wind turbine processing or probe processing steps could allow the model





to approximate a more complex operator. A mixed-modeling approach could also be developed, incorporating PyWake as an additional approximator alongside the GNO to predict a higher-fidelity flow field, thereby forming a multi-fidelity upscaling framework. Integrating this with a more advanced attention mechanism could enable the formation of more physically meaningful graph connections. Although such an approach would likely be more computationally expensive than the model proposed in this work, it would serve as a CFD surrogate and thus remain comparatively affordable.

Overall, the GNO presented in this work provides a robust baseline for efficient, data-driven flow prediction in complex wind farm environments. Further improvements can be achieved through the incorporation of higher-fidelity training data, enhanced attention mechanisms, and multi-fidelity coupling strategies, which have the potential to improve its predictive performance. As these developments are implemented, the GNO model may emerge as an essential tool for both research and industrial applications.



*Code and data availability.* The code used to train the GNO is publicly available at https://github.com/jenspeterschoeler/Wind-Farm-GNO. Owing to the large size of the whole dataset, only the test set is hosted on Zenodo (Schøler et al., 2025). However, the data can be re-generated using the code at https://github.com/jenspeterschoeler/Wind-farm-Graph-flow-data.





## Appendix A:  GNO Dataloading

As with other neural operators, the data comes in triplets, cf. branch input, trunk input, and target output. Because both the GNO branch and trunk are GNNs, the branch input is a graph, and the trunk is a second graph with a separate edge configuration. One of the core strengths of GNNs is the ability to process graphs of different sizes. However, to leverage the efficiency of the Jax framework, it is necessary to use JIT compilation, which requires inputs to have fixed sizes and shapes. To create graphs of fixed sizes and shapes, graphs are dynamically batched together until they reach a maximum size, defined by the number of graphs, total nodes, and total edges.

Batching graphs differ from conventional batching. In traditional batching, inputs share identical dimensions, and batching is performed along a new dimension. However, because the number of nodes and edges varies for each graph, batching is achieved by combining multiple graphs into a single larger graph. This approach preserves the sub-graphs individual characteristics by ensuring no edges between sub-graphs, thus preventing interactions during training and inference. Dynamic batching involves adjusting the number of graphs in each batch to fit within the size constraints rather than using a fixed batch size as in conventional batching. When dynamic batching is employed, adding another graph to the batch is prohibited once any maximum size limit is reached, as doing so would violate the constraints. Instead, padding with empty graphs, edges, and nodes is used to meet the target size.

An additional challenge arises because Jax does not currently offer a native data loader. At the time of writing, the most popular and mature Python framework for GNNs is PyG. Consequently, the PyG data loader has been used to load data in parallel from the file system into memory. Afterward, the graphs are converted to the Jraph format. Finally, dynamic batching and padding are performed sequentially on the CPU before the data is transferred to the GPU for processing. The current performance bottleneck lies in the sequential dynamic batching process. Parallelizing this step is challenging due to the variable sizes of the graphs. However, this method is still significantly more efficient than loading data from the file system sequentially. The process is visualized in Fig. A1.

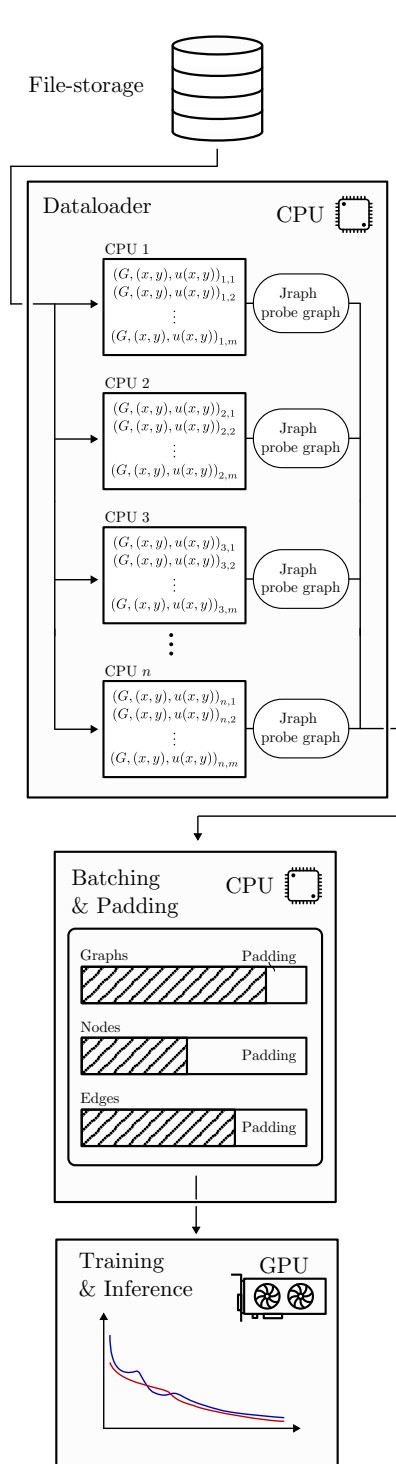

**Figure A1.** Flowchart diagram illustrating the steps from on-disk memory, constructing the Jraph compatible probe graph in parallel, batching, padding, and moving the resultant batched and padded probe graph to video memory.





*Author contributions.* JPSCH and PIRE conceived the research; JPSCH developed the methodology; JPSCH developed the code for the GNO and generated the required data; JPSCH performed the formal analysis; PIRE and JQ supervised the work; JPSCH, JQ, FPWR wrote the original draft; and JPSCH, JQ, FPWR and PIRE reviewed the draft.

*Competing interests.* The authors declare that they have no competing interests.

*Acknowledgements.* This work was partially funded by TotalEnergies under the "Inter-Farm Interactions (IFI)" project.

The authors thank Mikkel N. Schmidt at DTU Compute for providing valuable feedback on graph neural networks.

Model training was performed on the Sophia cluster (Technical University of Denmark, 2019), without which the GNO could not have been trained. The authors also acknowledge the use of AI tools in preparing this manuscript and the associated code. In particular, ChatGPT, Grammarly, and GitHub Copilot were used for proofreading and code auto-completion.





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
