# Peer review of "Graph Neural Operator for windfarm wake flow"

_Wind Energy Science, 2025_

## Referee Comment (RC1)

2025-12-12

The manuscript wes-2025-261 presents an advanced deep-learning approach for constructing a surrogate model to predict wind-speed deficits within and downstream of wind farms. The model is trained on a large data set generated by a low-fidelity, steady-state wake model (PyWake). The authors highlight prediction speed and generalisation to a range of wind-farm layouts and inflow conditions as key advantages. The deep-learning architecture is described in detail, and many test results are provided.

Although the model architecture is sophisticated and the method is thoroughly described, I believe the manuscript requires major revisions for the following reasons:

- The manuscript concludes that the proposed method represents a *breakthrough* in data-driven wind-farm modelling and provides a *robust baseline* for efficient flow prediction. However, the literature already contains ML-based surrogate models, including GNN-based and other approaches. The manuscript does not demonstrate that the proposed model outperforms these (simpler) existing alternatives or offers clear additional value. To substantiate the claimed contributions, a comparative analysis is needed.
- The performance metrics reported in the abstract and conclusion of the manuscript (RMSE and MAPE) are difficult to interpret (without having read in detail the complete manuscript) and, in my view, give an overly optimistic impression. As the *primary purpose* of PyWake—and by extension of the surrogate model—is to accurately represent wake flow velocity deficits, the MAPE should be defined with respect to the *ground-truth velocity deficit*, instead of to the free-stream wind speed. As shown in Figure 9, for moderate to high wind speeds, the predicted deficits in waked zones deviate by more than 100% from the ground-truth wind velocity deficits. In addition, the reported RMSE value is averaged over a large domain that may include large zones without any wind velocity deficit, where the prediction problem is trivial. This may inflate this performance metric and should be clarified or adjusted. In order to assess the accuracy of the proposed GNO approach, it may also be useful to compare its accuracy to the accuracy of PyWake itself (either relative to real-world data or high-fidelity models), as reported in previous studies.
- The abstract notes as motivation for implementing a surrogate model, the many required simulations for applications such as wind-farm layout optimisation and consideration of neighbouring wind farms. It is therefore recommended that the manuscript explicitly addresses whether—and by how much—the proposed surrogate model really adds value for these targeted applications, as well as any limitations that may arise from using the GNO-approach compared to using PyWake.
- Readers of Wind Energy Science are typically not AI specialists. The manuscript contains many low-level implementation details that obscure the core ideas and may discourage readers from completing the lecture of the manuscript. Therefore, consider moving purely technical implementation aspects to an appendix. As the software code is openly available (which is greatly appreciated), some details that are generic to deep-learning workflows might even be omitted entirely from the manuscript.

Below, I list more specific comments by section.

Abstract

Line 3: The text implies that the classical superposition principle is a disadvantage of existing approaches, yet the surrogate model is trained by a model that implements such an approach. Please rephrase to avoid suggesting a contradiction that is only clarified in the conclusion of the manuscript.

Line 8: "simulated wind farms": I recommend that, for clarity, it is mentioned explicitly that it concerns PyWake or, more generally, a low-fidelity steady-state engineering wake model.

Line 10: "underestimated … wake effects". For clarity, add: "compared to the simulated values".

**1. Introduction**

Line 61: Many WES readers will not be familiar with Graph Neural Operators (GNOs). Since GNNs are introduced earlier and better known, consider briefly describing how GNOs relate to GNNs and why they may yield improved performance.

Line 74: Use "Section" (capital S). In general, ensure consistent capitalisation and punctuation throughout the manuscript (e.g., figure 2, line 121, line 129, line 140, line 209, line 240, line 320, line 404, line 416, …)

**2. Methodology**

Line 78: "Here, … ". I suggest improving the formulation. (at multiple occasions in the manuscript)

Line 104: Alternative formatting: $R_0^+$

Line 120: Should appear on line 119.

Lines 121 – 126: Difficult to follow; please rephrase for clarity.

Line 130 - : This section presents extensive detail on selected model parameters. Clarify which parameters materially affect results. Non-essential implementation details may be moved to an appendix. Additionally, because not all readers are familiar with PyWake, please summarise its key assumptions (steady-state flow, homogeneous inflow, no yaw misalignment or curtailment, etc.). Also, clarify whether the same CT-curve is used for all simulations, and discuss implications for generalisation. (As you may know, examples in literature exist attempting to integrate the CT-curve as node features in the GNN).

Line 133: Citation formatting issue.

Line 133: "This model was chosen due to its relative simplicity". Does this have any influence on the generalisation of the results of the manuscript? Please make this explicit in the manuscript.

Line 135: "The authors fitted the model to a LES." Why is this relevant here?

Line 143: "…model…chosen for its simplicity". Does this have any influence on the generalisation of the results of the manuscript? Please make this explicit in the manuscript.

Line 145: When "TI" is used as a variable, the formatting shall be that of a variable. (This comment is applicable throughout the manuscript.)

Line 170: "A linear sum is used for the wake summations." Here, it could be stated/repeated that the GNO theoretically may also handle non-linear superposition, but this may be verified in future work.

Line 179: "The GNO is gird invariant." Begin a new paragraph.

Line 180: "… from flow away from …" ?

Line 186: $\bar{x}$ could be interpreted as a vector or maximum; consider using: $x'$

Figure 3: Clarify why the coordinate origin differs from the farm centre. How the coordinate origin is determined? Does this influence (the features of) the GNO?

Lines 191- 193: Appear irrelevant; consider removing.

Line 198: "Instead, they are copied to each node as node features." Why did you make this decision? provide references if applicable.

Line 206: Consider adding $V = V_{wt} \cup V_p$.

Line 209: Consider adding the analogous equation to Eq. 7.

Line 209: "The inter-turbine edges…": start a new paragraph for readability.

Line 204: "Probe node…": start a new paragraph for readability.

Line 216: "… it uses Delaunay triangulation…" This is repeated in line 220. Consider rephrasing.

Line 224: You state that Cartesian coordinates are used as edge features (and not as node features, which intuitively would make more sense), which seems to contradict later text. Please clarify and justify the design choice for the node and edge features; add references if similar implementations exist. Also specify how the coordinate origin is chosen, as it has an influence on the Cartesian coordinates.

Lines 230 – 235: The sequence and interaction of the models are unclear; Consider rephrasing.

Line 239: Clarify "common" and "abstraction".

Line 240: "initial two stages":  Clarify whether these are the encoder and approximator?

Line 245: "sped up"

Line 248: "Fore instance, in fully couple formulations, …": is this the case in your implementation?

Lines 256 – 260: This information deserves a separate subsection somewhere else in the manuscript.

Line 267: "Q": here a capital letter, in contrast to in line 278.

Line 278: The equation comes back (and better) in lines 292 – 294

Equation (10a): k = 1,…,K; Move be on the same line.

Lines 292-294: For clarity, repeat here the numeric values in your implementation for |E|, fe, …, and Q.

Line 300 and equation 12b: Shouldn't this be $\phi_{h'}$ ?  (ref. notation in Figure 5)

Line 309: Define $\widehat{x}_j$ .

Figure 7: Nothing appears marked yellow.

Equation 18b: To be a relevant performance measure for accuracy, the MAPE of the predicted wind speed deficit should compare the predicted wind speed deficit relative to the ground-truth wind speed deficit (instead of comparing to the free inflow wind speed).

Line 371: "limiting factor has been the 72-hour walltime": Was the performance the GNO still improving at the end of this walltime? If yes, why could the model not be saved to resume the training of the model, starting a new walltime?

Line 385: Grammar issue.

Line 397: Please explain more clearly.

Line 400: "To evaluate the model under realistic conditions." This suggests that the other layouts would not be realistic. Consider rephrasing.

Line 401: "farm layout"

Line 405: "simple operation": Does the applied power curve not depend on TI? If yes, how is TI at the turbine locations predicted by the GNO?

3. Results and Discussion

Line 419: "internal MLP": Clarify what is meant by "internal".

Line 229: "probes per graph": how are these selected? Randomly in the large box around the wind farm?

Line 429: "number of batched graphs": Do you mean, the number of graphs per batch?

Line 436: "increases the confidence that the right model is chosen": Why?

Table 2: "lowest best metrics marked with b": What is meant by that?

Table 2: Add the unit of each performance indicator.

Table 3: Add the unit of each performance indicator.

Table 3: The values of the performance indicators for the test set seem to be of another order of magnitude compared to those of the validation set. How is that possible?

Table 3: Consider adding baseline metrics for naïve deficit profile(s) (such as the zero-deficit profile), so that the magnitude of the metrics for the GNO models can be better interpreted.

Line 455: Include a wake profile(s) *inside* the farm (i.e., for $\bar{x} < 0$), so that not only the far-wake profile is analysed, but also the near-wake modelling is analysed.

Line 461: Similar to one of the main comments: What is the range of the ground-truth velocity deficits relative to U? This may also be small.

Figure 9: The plots are too small to be readable on a printed version of the manuscript.

Figure 9: Comparing the close results of dashed and full lines with different colours (and thus different wind speeds), at first view, one could think that there has been a failure with the allocation of the colours.

Figure 10: It would be interesting to see the plots for the 3 wind speeds separately, or for the wind speed 12ms-1 only (i.e. wind speed with the highest wake losses), in case of lack of space (or in the appendix).

Figure 11: It looks as if the orientation of the wake is rotated from the main axes of the farm layouts in Figure 8.

Figure 11: Make plots larger to be readable in printed A4 version.

Line 481: "conditions similar to their training data": Are you suggesting that the optimised layout is less similar to your training data? If that were known, it would be recommendable to add better layouts to your training data.

Line 485: In order to validate whether the accuracy of the proposed method is sufficient for optimisation applications (which is the stated motivation for implementing this GNO model), it would be very interesting to compare the AEP predictions made on the basis of PyWake and the GNO, for both wind farm layouts.

Line 485: For (farm layout) optimisation problems, the absolute value of the wake losses may be less important than the relative difference between for different layouts. Therefore, consider adding measures about the relative wake power losses predicted by the GNO.

Line 530: I understand your choice to compare computational cost on a one-core CPU. Nevertheless, as currently the GNO can be run on 32 cores simultaneously and on a GPU, in contrast to PyWake, in my opinion, you may also add this comparison in the manuscript. Indeed, that would be the reality for someone who may have to choose between PyWake and the surrogate model.

4. Conclusion

Line 553: "fresh". Consider alternative wording.

Line 555: 'afterthought", "established literature": What is meant here?

Line 559: "In section 3": leave away

Line 562: "The GNO is less accurate that PyWake." You did not prove that, but it is trivial.

See main comments.

Appendix A

Line 601: "cf. branch input, trunk input": Please clarify these terms.

---

## Referee Comment (RC2)

The authors introduce a Graph Neural Operator architecture for wind farm wake flow prediction, where turbine-turbine interactions are learned through message-passing on a graph and flow velocities can be queried at arbitrary probe locations. Trained on ~21,000 PyWake simulations spanning diverse procedurally-generated layouts, the model generalizes across farm configurations and seems to achieve low RMSE in the predictions while being approximately 10x faster than the engineering baseline (PyWake).

In general, I find that the manuscript is well written and comprehensive: all components of the framework, from procedural dataset generation to GNO architecture and training, are described in sufficient detail to enable independent implementation. The breakdown of the results is also very detailed, and I appreciate the thorough reporting on computational aspects. I recommend minor revisions, with specific comments below.

**Specific comments:**

- Section 2
    - A lot of detail is provided for the models that are used by PyWake to generate the training data. I think some of this can be moved to the appendix.
    - The part about the additional coordinate system is a bit confusing and could do with some extra clarification.
    - Why did you choose to use the coordinates of the sender nodes as an edge feature, instead of the relative coordinates (vector coordinates) between the the two connected nodes, as is typically done in other GNN work?
    - I'm not sure I see the advantage of having the decoding stage setup as it is. I get that you want to first freeze the turbine node latents, so that you have the option of adding any amount of probe nodes after, but why not have undirected probe edges between a probe and other probes, as well as between a probe and the turbine nodes. I guess the current setup makes sense when only using a single probe message-passing step, but why not have multiple steps during this phase? It would increase performance (albeit with extra computational costs). Please consider clarifying this.
    - Could you have not resumed training after 72 hours? Or were the models sufficiently trained by this time and it was not needed?
    - I'm not sure the MAPE metric is correct, as typically we would divide the error by the target value. In some sense, what you show here is more of a MAE that is normalized by the inflow velocity, which may still be a useful metric.
    - You mention the cardinality of the graph, is this actually used somewhere?
- Section 3
    - You could consider moving some of the parameter tables for the hyperparameter study to the appendix.

- No ablation on the number of message-passing steps (M=3 throughout) is performed. This seems like an important hyperparameter to study, given that message-passing propagates information only to immediate neighbors. Would increasing the number of steps increase performance for cluster farm setups, where wake superposition seems to be the biggest hurdle?
- The training procedure regarding probe node sampling deserves clarification. How are the probe locations selected? Random uniform sampling across the domain, or weighted toward regions with stronger wake effects? This choice likely influences model performance in different flow regions.
- I would have really liked to see a complete flow map comparison between the GNO's output vs PyWake (along with a difference plot), especially in the near-wake region. It would be extremely useful to understand where the model falls short. You could do this for the IEA Wind 740-10-MW reference reference farms for instance.
- This point is optional, but I was wondering what the learned RBF kernels look like. Showing the learned RBF distributions after training would provide insight into what spatial scales the model finds important.
- Figure 9. I think that you can plot the farm in a nicer way, it is hard to actually see the positions of the turbines.
- Figure 12. In general I find this plot a bit hard to read, the color scheme can be improved and the bars made a bit wider. For subplot (a): the KDE extends into negative RMSE values, which is not physically meaningful. Consider using a boundary-corrected KDE method, or simply truncating/reflecting at zero.
- Section 4
  - I would suggest more measured language that emphasizes the nice methodological contribution without overclaiming impact ('breakthrough').

Minor technical comments:

- Line 120, line 283: alignment issues
- Line 320: text overflow
- "Windfarm" vs "Wind farm" used interchangeably, try to stay consistent. Same for "flowmap" vs "flow map".
- Caption of Table 2: typo in caption ('b' instead of 'bold'?)